# Trajectories of childhood immune development and respiratory health relevant to asthma and allergy

Howard HF Tang[1,2]*, Shu Mei Teo[1,3], Danielle CM Belgrave[4], Michael D Evans[3,5], Daniel J Jackson[5], Marta Brozynska[1,4], Merci MH Kusel[6], Sebastian L Johnston[7], James E Gern[5], Robert F Lemanske[5], Angela Simpson[8], Adnan Custovic[4], Peter D Sly[6,9], Patrick G Holt[6,9], Kathryn E Holt[10,11], Michael Inouye[1,3,12]*

[1]Cambridge Baker Systems Genomics Initiative, Baker Heart and Diabetes Institute, Victoria, Australia; [2]School of BioSciences, The University of Melbourne, Victoria, Australia; [3]Cambridge Baker Systems Genomics Initiative, Department of Public Health and Primary Care, University of Cambridge, Cambridge, United Kingdom; [4]Department of Paediatrics, Imperial College London, London, United Kingdom; [5]University of Wisconsin School of Medicine and Public Health, Madison, United States; [6]Telethon Kids Institute, University of Western Australia, Perth, Australia; [7]Airway Disease Infection Section, MRC & Asthma UK Centre in Allergic Mechanisms of Asthma, National Heart and Lung Institute, Imperial College London, London, United Kingdom; [8]Division of Infection, Immunity and Respiratory Medicine, The University of Manchester, Manchester, United Kingdom; [9]Child Health Research Centre, The University of Queensland, Brisbane, Australia; [10]Bio21 Molecular Science and Biotechnology Institute, The University of Melbourne, Victoria, Australia; [11]The London School of Hygiene and Tropical Medicine, London, United Kingdom; [12]The Alan Turing Institute, London, United Kingdom

*For correspondence:
Howard.Tang@baker.edu.au
(HHFT);
mi336@medschl.cam.ac.uk (MI)

Competing interests: The authors declare that no competing interests exist.

**Abstract** Events in early life contribute to subsequent risk of asthma; however, the causes and trajectories of childhood wheeze are heterogeneous and do not always result in asthma. Similarly, not all atopic individuals develop wheeze, and vice versa. The reasons for these differences are unclear. Using unsupervised model-based cluster analysis, we identified latent clusters within a prospective birth cohort with deep immunological and respiratory phenotyping. We characterised each cluster in terms of immunological profile and disease risk, and replicated our results in external cohorts from the UK and USA. We discovered three distinct trajectories, one of which is a high-risk 'atopic' cluster with increased propensity for allergic diseases throughout childhood. Atopy contributes varyingly to later wheeze depending on cluster membership. Our findings demonstrate the utility of unsupervised analysis in elucidating heterogeneity in asthma pathogenesis and provide a foundation for improving management and prevention of childhood asthma.
DOI: https://doi.org/10.7554/eLife.35856.001

## Introduction

Asthma is a global health problem, and there is a pressing need for better understanding of its pathogenesis (*Global Initiative for Asthma, 2015*). Asthma is strongly associated with allergy, and both genetic and environmental factors may be involved (*Ober and Yao, 2011*; *Dick et al., 2014*). The 'hygiene hypothesis' proposes that modern changes to hygiene, sanitation and living environment

**eLife digest** Asthma causes wheezy and troubled breathing, and can be life-threatening. Scientists and doctors understand that asthma begins in early childhood. Chest infections, exposure to bacteria, viruses, and allergies may cause or trigger asthma. One person with asthma may not have the same origins as another. But it is not yet clear how various triggers may interact to trigger or exacerbate asthma.

To disentangle how these factors contribute to asthma, experts have tried to group people with asthma into subgroups. Unfortunately, the groups often vary from expert to expert. Now, some scientists are using computers to sort patients with asthma. The scientists let the computers decide the best criteria for sorting patients. This way the machines may identify patterns that are not obvious to humans.

Using this computer-based approach, Tang et al. sorted Australian children with asthma into 3 groups based on their early life allergies and respiratory health. One group has high-risk asthma with frequent chest infections and strong allergic responses. The other two groups are low-risk, but they respond differently to allergy and infection. Common tests used by doctors to diagnose patients with allergy or asthma may not work the same with all three groups. The bacteria found in the nose influence the risk of asthma, even in patients who are well, and the way this occurs varies by group. Similar groups were also found among children with asthma in the United States and the United Kingdom.

Learning more about subgroups of patients with asthma may help other scientists and doctors design better ways to diagnose, treat, or prevent asthma. Working together with scientists around the world to determine how to best describe subgroups of people according to asthma type and risk is a critical step in the process. Tang et al. hope other scientist will test whether these three groups are also found in people from other parts of the world.

DOI: https://doi.org/10.7554/eLife.35856.002

have modified human exposures to microbes, with subsequent effects on early-life immune development (*Okada et al., 2010*). However, the clinical presentation and prognosis of childhood wheeze is highly variable: some children remit; others remit but relapse; and yet others have wheeze persisting into adult asthma (*Morgan et al., 2005*). These differences suggest that the underlying causes of disease also differ from person to person. For example, while asthma is commonly linked to allergy, not all individuals with wheeze are sensitised to allergen, and vice versa (*Spycher et al., 2010*). As such, childhood asthma is a heterogeneous condition (*Hekking and Bel, 2014*; *Wenzel, 2012*), and this greatly complicates the study of its pathogenesis (*Anderson, 2008*). We postulate that there are subpopulations in early childhood, each sharing similar patterns of pathophysiology, disease susceptibility and phenotype that permit categorisation into clusters. If we can agnostically identify these clusters, then we may explore the biological mechanisms that underlie them, and find targets for early intervention that are specific for different asthma subtypes.

Previous attempts at subtyping asthma susceptibility relied on supervised classification, using expert knowledge and cut-offs to define clusters. For example, criteria such as – specific immunoglobulin E (IgE) $\geq$0.35 kU/L; wheal diameter $\geq$3 mm in a skin prick test (SPT); or symptom score surpassing a threshold –may determine classification into a high-risk profile (*Castro-Rodríguez et al., 2000*; *Frith et al., 2011*). However, these cut-offs vary with age, gender or other parameters, and may not accurately reflect true attribution of risk (*Linden et al., 2011*). Hence, they often continue to produce heterogeneous groups. Furthermore, previous studies tended to focus on a single 'domain', for instance grouping only by immunological response (*Prescott et al., 1999*), symptomatology or timing of disease (*Martinez et al., 1995*; *Kurukulaaratchy et al., 2003*). Recently, researchers have turned to unsupervised approaches, such as model-based cluster analysis and latent class analysis (LCA) (*Deliu et al., 2016*; *Lazic et al., 2013*; *Simpson et al., 2010*; *Belgrave et al., 2014*; *Belgrave et al., 2013*; *Wu et al., 2015*). These do not require experts to supply cut-offs, but can instead 'learn' boundaries from the data. They can potentially uncover patterns of similarity not immediately obvious to the human eye. Finally, these methods can cover a broader

range of domains, incorporating measurements from multiple sources to determine clusters that are potentially informative of asthma risk.

Here, we use a data-driven unsupervised framework together with a comprehensively phenotyped birth cohort, to define developmental trajectories during preschool years, a period known to be critical to asthma pathogenesis. Specifically, we (1) use non-parametric mixture models to discover latent clusters that define early childhood trajectories of immune function and susceptibility to respiratory infection; (2) investigate how these clusters relate to differential profiles of asthma susceptibility, and to existing definitions of atopy; (3) identify risk factors for asthma within each cluster; and (4) externally validate the clusters in independent cohorts.

## Results

To characterise the broad structure of an Australian dataset of young children (Childhood Asthma Study, CAS), we performed principal components analysis (*Figure 1—figure supplement 1*). Afterwards, to explicitly model the heterogeneous data types of the cohorts as well as explicitly identify clusters, we used non-parametric expectation-maximiation (npEM) mixture models (Materials and methods). By applying npEM-based clustering and classification to CAS, we identified three distinct clusters from 217 individuals and 174 clustering features (*Figure 1*): low-risk CAS1 (*N* = 88, 25% wheeze at age 5), low-risk but allergy-susceptible CAS2 (*N* = 107, 21% wheeze at age 5) and high-risk CAS3 (*N* = 22, 76% wheeze at age 5). Forty-six individuals in CAS had excessive missing data and were not classifiable. The CAS clusters satisfied basic measures of internal stability and were distinguishable on a PCA plot of the complete-case dataset (*Figure 1—figure supplement 1*). A graphical summary of results for the CAS clusters is presented in *Figure 2*.

### CAS1: low-risk, non-atopic cluster with transient wheeze

CAS1 was a low-risk cluster with infrequent and transient respiratory wheeze. Rates of wheeze declined from 33% at age 1% to 12% by age 10 (*Table 1*; *Figure 3*). In this cluster, Th2 cytokine responses of peripheral blood mononuclear cells (PBMCs) to allergen stimulation were minimal; and rates of allergen sensitisation (as measured by IgE or skin prick test, SPT) were the lowest among all groups (*Table 2*; *Figure 4*; *Supplementary file 1* – table supplement 3B-D). IgG and IgG4 were also low across all allergens.

Frequency of respiratory infection in CAS1 was low (*Table 3*). However, high frequency of lower respiratory infections (LRIs) in childhood, especially wheezy LRIs (wLRIs), was a risk factor for age-5 wheeze – even after adjusting for sex, body mass index (BMI) and parental history of asthma as demographic covariates (*Table 4*). Repeated-measures ANOVA identified that LRI and wLRI frequency in the first 3 years were predictors for age-5 wheeze (*Supplementary file 1* – table supplement 4); however, timepoint-specific analyses showed that differences were only noticeable from age 3 onwards (*Table 4*; *Figure 5A–B*). A multiple regression model with stepwise elimination yielded three significant variables: age-three wLRI frequency (odds ratio OR 5.6 per unit increase, p=0.0068); age-four LRI frequency (OR 3.6, p=0.018); and a protective effect from proportion of infection-associated microbiome profile groups (MPGs; *Streptococcus*, *Haemophilus*, *Moraxella*) in age-two-to-four healthy nasopharyngeal aspirate samples (NPAs; OR 0.19 per quartile, p=0.014).

### CAS2: low-risk cluster susceptible to atopic and non-atopic wheeze

Similar to CAS1, CAS2 was a low-risk cluster with infrequent allergic disease. Compared to CAS1, Phadiatop and house dust mite (HDM) IgE were elevated at most timepoints (*Table 2*; *Figure 4A*; *Supplementary file 1* – table supplement 3B), with the exception of peanut IgE (Wilcoxon, adjusted p=0.99 at all timepoints; *Figure 4D*). CAS2 IgG and IgG4 were intermediate between CAS1 and CAS3 levels; CAS2 IgG was closer to CAS1, while CAS2 IgG4 was closer to CAS3 (*Table 2*; *Figure 4*). Despite these antibody differences, yearly rates of wheeze in CAS2 remained comparable to CAS1 (30% at age 1, declining to 18% at age 10; *Table 1*; *Figure 3*). Interestingly, compared to CAS1, individuals in CAS2 had fewer older siblings living in the household at age 2, as well as more frequent paternal history of asthma (adjusted p=0.029 and 0.055, respectively; *Supplementary file 1* – table supplement 3A).

Predictive factors for age-5 wheeze in CAS2 included: LRI, wLRI and febrile LRI (fLRI) frequency (GLM; p=$2.7 \times 10^{-3}$, 0.016 and 0.02 at age 3, respectively); HDM IgE (p=0.016 and 0.011 at ages 2

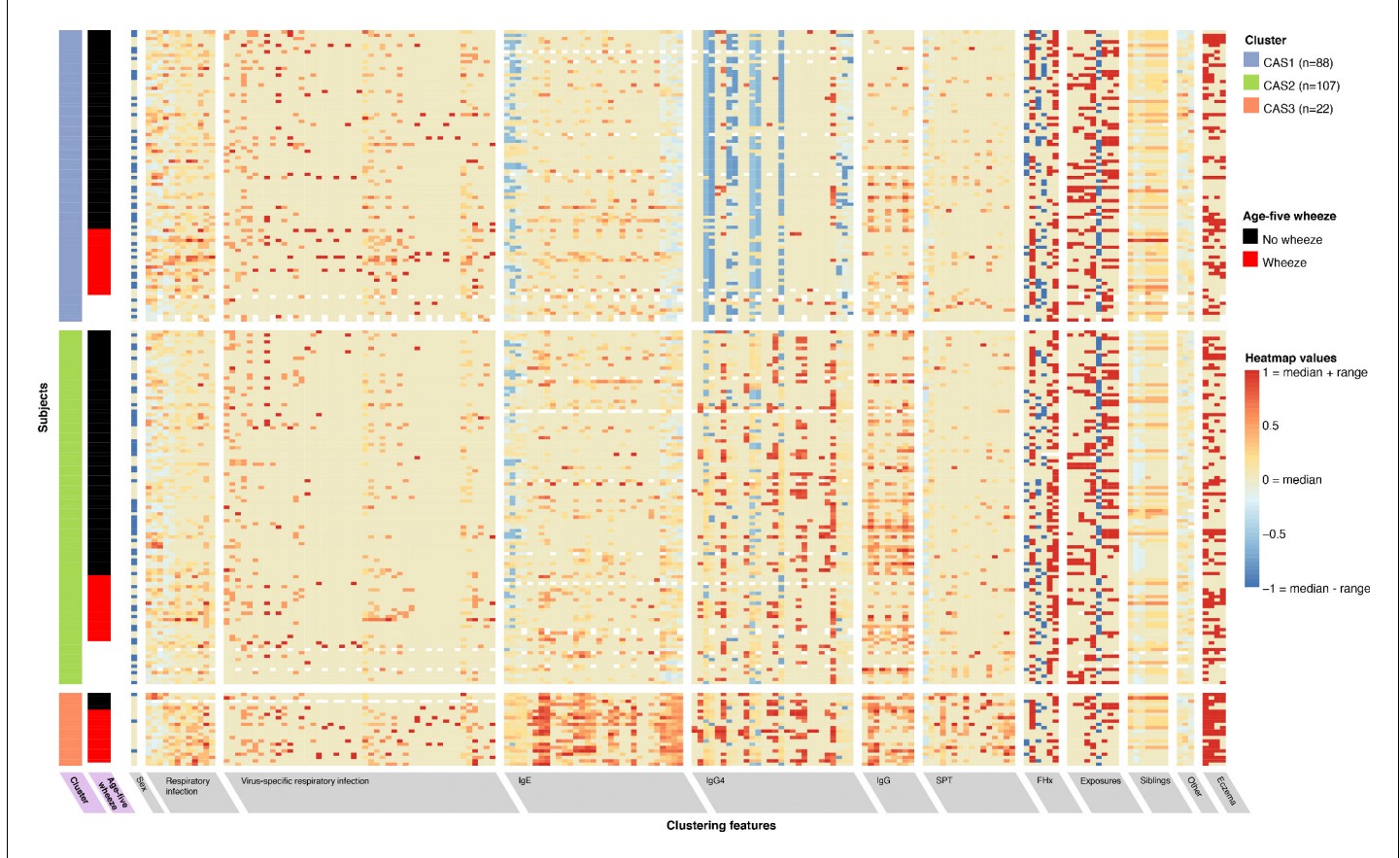

**Figure 1.** Non-parametric mixture-model-based clustering of CAS dataset, based on 174 features. SPT = skin prick test. White spaces within the heatmap indicate missing data. Rows represent individuals; columns represent clustering features with general categories as labelled on grey background. Variables with grey background are clustering features ordered by category or type of variable first (e.g. all HDM IgE-related variables grouped together), then by timepoint (earlier to later, from left to right). Variables with lilac background indicate resultant cluster membership and outcome variable (age-5 wheeze). Heatmap values are scaled relative to range and median values for each feature; the median is coloured beige-yellow, the median +range red, and median – range blue. For sex, −1/blue = female, 0/yellow (median) = male.
DOI: https://doi.org/10.7554/eLife.35856.003

The following figure supplements are available for figure 1:

**Figure supplement 1.** Scatterplot of principal components analysis (PCA) of the complete-case CAS dataset (*N* = 186), with points coloured by npEM clusters Each point represents an individual.
DOI: https://doi.org/10.7554/eLife.35856.004

**Figure supplement 2.** Silhouette widths of clusters generated by npEM.
DOI: https://doi.org/10.7554/eLife.35856.005

**Figure supplement 3.** Overview of study methodology.
DOI: https://doi.org/10.7554/eLife.35856.006

and 4, respectively); and Phadiatop IgE (p=0.01 at age 4) (*Table 4*). Repeated-measures ANOVA showed that HDM IgE and LRI-related variables (LRI, wLRI, fLRI) from the first 3 years were significant predictors of age-5 wheeze (*Supplementary file 1* – table supplement 4). Timepoint-specific analyses showed that differences were observable in HDM IgE and fLRI from age 2 onwards, while in LRI and wLRI they were only noticeable from age 3 (*Table 4*; *Figure 5*). A multiple regression model with stepwise elimination identified three significant variables: age-2 fLRI (OR eight per unit increase, p=0.0075), age-4 wLRI (OR 5.3 p=0.0016), and age-4 Phadiatop IgE (OR 3.3, p=0.0088). But although both IgE-related and infection-related risk factors contributed to age-5 wheeze, there was no significant evidence of interaction between them (p=0.36 within CAS2 alone, p=0.92 across entire cohort, for age-4 wLRI frequency ×Phadiatop IgE). Overall, CAS2 represented a low-risk trajectory susceptible to, but not necessarily afflicted by, wheeze due to atopic and non-atopic risk factors. In

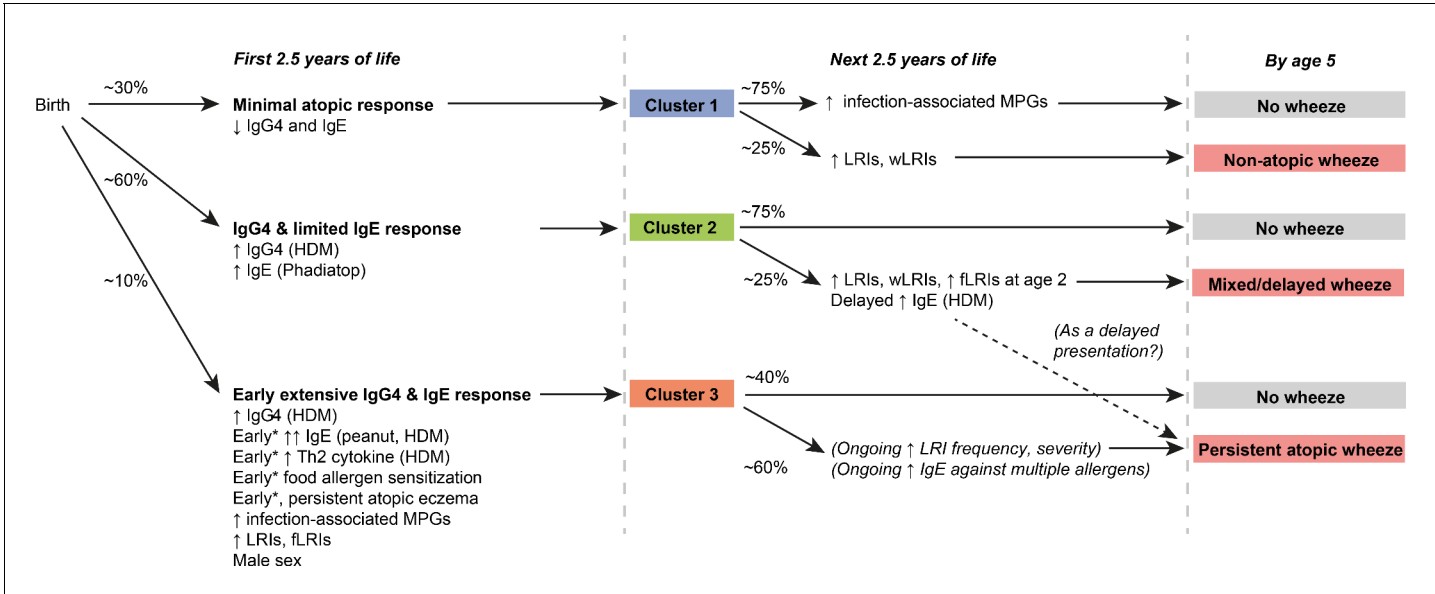

**Figure 2.** Graphical summary of proposed clusters *'Early' specifically refers to 'within the first 6 months of life'.
DOI: https://doi.org/10.7554/eLife.35856.007

this cluster, atopic determinants of age-5 wheeze were only active from age 2 onwards, suggesting delayed atopic wheeze in this cluster. This duality of atopic and non-atopic risk factors for wheeze in this cluster was further supported by decision tree analysis, which identified that wheezy LRI frequency and HDM IgE best separated wheezers from non-wheezers in CAS2 (*Figure 5—figure supplement 3*).

## CAS3: high-risk atopic cluster with persistent wheeze

CAS3 was a 'high-risk' cluster, where persistent respiratory wheeze and atopic disease was seen in more than half the group throughout the first 10 years of life (*Table 1*; *Figure 3*). This cluster was dominated by males (86%, Fisher exact test, unadjusted $p=6.8 \times 10^{-3}$ compared to CAS1, *Table 1*), and appeared to represent an early- and multi-sensitised atopic phenotype with persistent wheeze. CAS3 had elevated IgE, IgG, and IgG4 responses to common allergens, especially Phadiatop, HDM and peanut IgE from 6 months onwards (*Table 2*; *Figure 4*; *Supplementary file 1* – table supplement 3B). SPTs were also more frequently positive in CAS3, especially to HDM and food allergens (peanut, cow's milk and egg white, *Supplementary file 1* – table supplement 3D).

No strong predictors for age-5 wheeze were identified within CAS3 (*Table 4*): only couch grass IgE at age 2 and acute respiratory infection (ARI) frequency at age 1 were weakly significant (both p=0.046). Neither of these reached statistical significance when incorporated in the same model. However, the prolific IgE response, and the frequency and severity of early-life LRIs in this cluster (*Table 3*), strongly suggest contribution from both atopic and non-atopic causes of wheeze. Hence, CAS3 primarily represented those with extreme levels of atopic sensitisation and infection. The relative paucity of identifiable predictors may be explained by the small size of CAS3 (N = 22), the intrinsically high rate of wheeze in the cluster (76% with age-5 wheeze), and saturation of risk from high levels of IgE and frequent infections.

Unlike the antibody measurements, cytokine measurements were excluded as clustering features due to high missingness. Nonetheless, with post-hoc analyses, we found that in vitro stimulation of PBMCs with HDM antigen elicited stronger Th2 cytokine responses in CAS3 compared to other clusters (*Table 2*, *Figure 6*). These cytokines (IL-4, IL-5, IL-13) were elevated from a very young age (Wilcoxon, adjusted $p=4.6 \times 10^{-5}$ for IL-4 mRNA at age 6 m, compared to CAS1), coinciding with increase in HDM IgE and IgG4 responses. Weaker but similar differences were observed for peanut- and ovalbumin-stimulated PBMCs at 6 months (unadjusted p<0.05 for all, *Supplementary file 1* –

**Table 1.** Comparison of selected demographic and clinical variables in CAS clusters

| Variable | Age (y) | Cas1 (N = 88) | Cas2 (N = 107) | Cas3 (N = 22) | P-value (unadjusted) | | | | Feature? |
|---|---|---|---|---|---|---|---|---|---|
| | | | | | Overall | Cas1 vs. 2 | Cas1 vs. 3 | Cas2 vs. 3 | |
| | | Prop. (95% CI) | Prop. (95% CI) | Prop. (95% CI) | Overall | Cas1 vs. 2 | Cas1 vs. 3 | Cas2 vs. 3 | |
| Sex = male | | 55% (44–65%) | 51% (42–61%) | 86% (71–100%) | 7.3E-03 | 0.67 | 6.8E-03 | 3.7E-03 | Yes |
| Maternal asthma | | 51% (40–62%) | 41% (32–51%) | 59% (37–81%) | 0.19 | 0.19 | 0.63 | 0.16 | Yes |
| Paternal asthma | | 22% (13–30%) | 44% (35–54%) | 23% (3.7–42%) | 2.2E-03 | 1.3E-03 | 1 | *0.093* | Yes |
| Wheeze | 1 | 33% (23–43%) | 30% (21–39%) | 55% (32–77%) | *0.092* | 0.76 | *0.084* | 0.046 | No |
| | 5 | 25% (15–35%) | 21% (13–30%) | 76% (56–96%) | 7.1E-06 | 0.59 | 2.6E-05 | 3.4E-06 | No |
| | 10 | 12% (3.4–21%) | 18% (8.4–27%) | 50% (24–76%) | 3.1E-03 | 0.46 | 1.5E-03 | 0.011 | No |
| Asthma | 5 | 15% (7–23%) | 13% (5.9–20%) | 52% (29–76%) | 4.1E-04 | 0.83 | 7.7E-04 | 2.1E-04 | No |
| | 10 | 10% (2.3–18%) | 15% (6.1–23%) | 56% (30–81%) | 2.6E-04 | 0.59 | 1.8E-04 | 7.9E-04 | No |
| Eczema | 6m | 39% (28–49%) | 45% (35–54%) | 91% (78–100%) | 2.4E-05 | 0.47 | 7.9E-06 | 9.0E-05 | Yes |
| | 1 | 34% (24–44%) | 30% (21–39%) | 82% (64–99%) | 2.5E-05 | 0.54 | 7.2E-05 | 1.4E-05 | Yes |
| | 5 | 28% (18–37%) | 24% (16–33%) | 71% (50–92%) | 2.1E-04 | 0.73 | 3.3E-04 | 7.9E-05 | No |
| Atopic rhinoconjunctivitis | 5 | 30% (20–40%) | 39% (29–49%) | 76% (56–96%) | 6.4E-04 | 0.21 | 2.7E-04 | 3.2E-03 | No |
| | | Mean (95% CI) | Mean (95% CI) | Mean (95% CI) | Overall | Cas1 vs. 2 | Cas1 vs. 3 | Cas2 vs. 3 | |
| BMI (kg/m$^2$) | 3 | 16 (16–17) | 16 (16–17) | 16 (16–17) | 0.86 | 0.65 | 0.68 | 0.8 | No* |
| | 4 | 16 (16–17) | 16 (16–16) | 17 (16–17) | 0.59 | 0.76 | 0.32 | 0.39 | No |
| | 5 | 16 (16–16) | 16 (16–16) | 16 (15–17) | 0.71 | 0.56 | 0.48 | 0.67 | No |
| | 10 | 18 (17–19) | 18 (17–18) | 18 (17–19) | 0.89 | 0.75 | 1 | 0.62 | No |
| Number of older siblings | 0 | 0.93 (0.72–1.1) | 0.53 (0.38–0.69) | 0.77 (0.32–1.2) | 4.5E-03 | 1.0E-03 | 0.37 | 0.25 | Yes |
| | 2 | 0.85 (0.66–1) | 0.5 (0.34–0.65) | 0.77 (0.32–1.2) | 2.8E-03 | 6.5E-04 | 0.48 | 0.16 | Yes |
| | 5 | 0.68 (0.5–0.85) | 0.39 (0.25–0.54) | 0.67 (0.23–1.1) | 0.016 | 5.1E-03 | 0.75 | 0.12 | No |
| | | Geom. mean (95% CI) | Geom. mean (95% CI) | Geom. mean (95% CI) | Overall | Cas1 vs. 2 | Cas1 vs. 3 | Cas2 vs. 3 | |
| Vitamin D (nmol/L) | 1 | 60 (55–64) | 59 (55–63) | 59 (52–67) | 0.93 | 0.98 | 0.76 | 0.7 | No |
| | 2 | 57 (54–61) | 58 (55–61) | 47 (40–55) | 0.012 | 0.82 | 5.4E-03 | 4.4E-03 | No |
| | 5 | 89 (83–95) | 84 (79–89) | 77 (69–84) | *0.057* | 0.46 | 0.016 | *0.056* | No |

BMI = body mass index; feature?=whether variable was used as a clustering feature or not; geom. mean = geometric mean; prop. = proportion. For categorical variables, associations were tested using Fisher exact test; for continuous variables, Kruskal-Wallis and Mann-Whitney-Wilcoxon. Bold text indicates statistical significance (p<0.05); italics indicate near-significance (p<0.10). *Not used as clustering feature, as BMI is a derived variable. Height and weight at age three were used instead.

DOI: https://doi.org/10.7554/eLife.35856.013

table supplement 3C). There were no other significant differences for other non-Th2 cytokines (IFN-γ, IL-10), nor were there specific differences for CAS1 or CAS2.

## Comparison of measures of immunological response

Across all clusters, allergen-specific IgG4 and IgG were positively correlated with IgE for the same allergen (especially HDM, *Figure 4—figure supplement 1*). As noted previously, CAS2 and CAS3 were distinguished from CAS1 by high IgG4, and CAS3 had greater IgG4 than either CAS1 or CAS2 (*Supplementary file 1* – table supplement 3B; *Figure 4*). Decision tree analysis (*Figure 5—figure supplement 1 to 3*) confirmed that IgG4-type variables strongly separated CAS2 and CAS3 from CAS1, while IgE-type variables separated CAS3 from the others.

Although previous literature suggests possible protection conferred by IgG4 (*Okamoto et al., 2012*) or IgG (*Holt et al., 2016*), in this study there was no clear evidence of such protection against

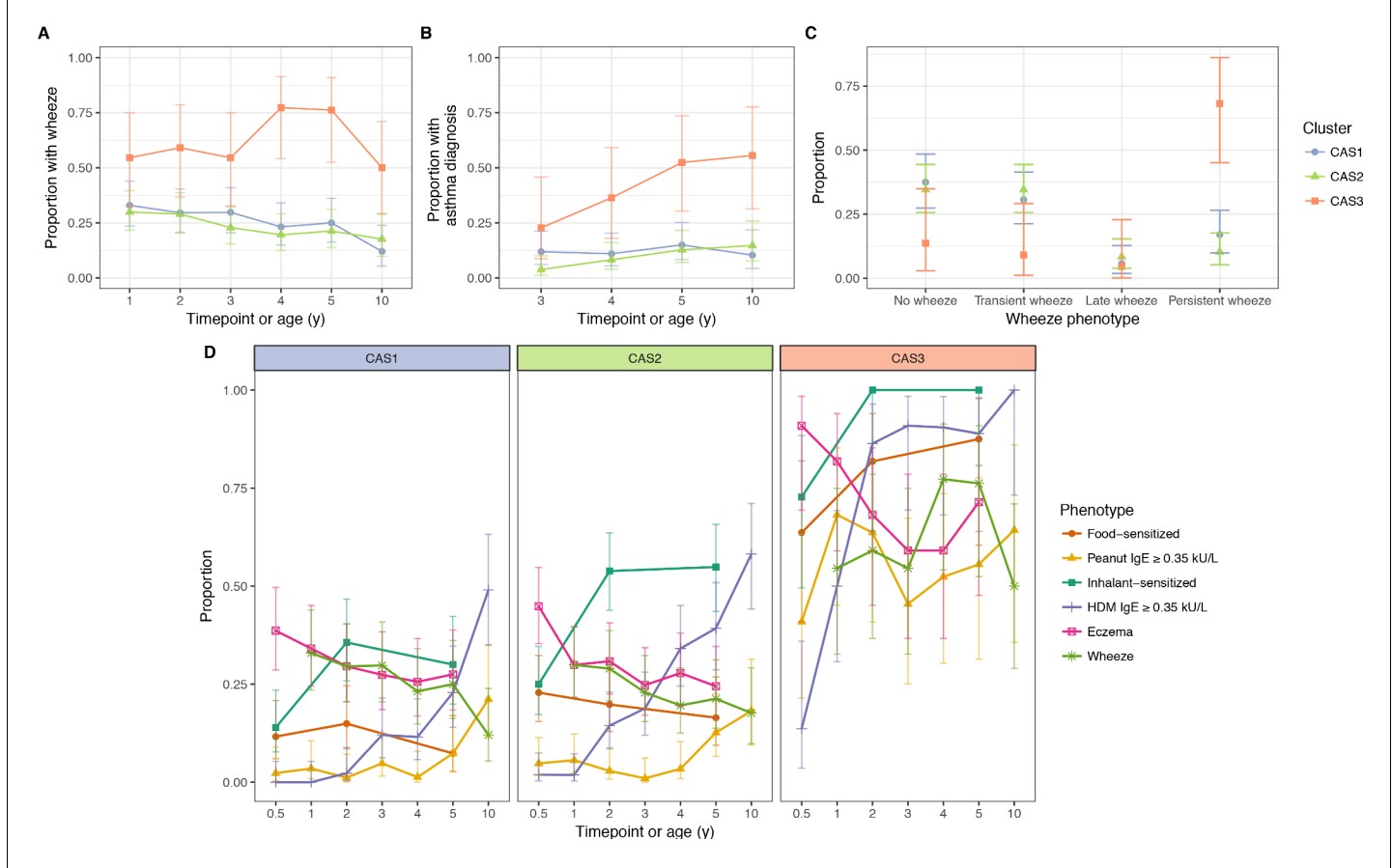

**Figure 3.** Incidence of multiple phenotypes, including parent-reported wheeze. (A) Physician-diagnosed asthma (B) defined wheeze phenotypes (C) in relation to food and inhalant sensitisation (D) stratified by cluster and time in the CAS dataset. Points indicate observed proportion; bars indicate 95% CI (binomial distribution). Wheeze phenotypes defined as: no wheeze = no wheeze at ages 1 to 3, or age 5; transient wheeze = any wheeze at ages 1 to 3, but not age 5; late wheeze = wheeze at age 5, but not ages 1 to 3; persistent wheeze = any wheeze at both ages 1 to 3 and age 5. Food sensitisation defined as peanut IgE ≥0.35 kU/L at any age, or cow's milk, egg white, peanut SPT > 2 or 3 mm for age ≤2 or>2 respectively. Inhalant sensitisation defined as HDM, cat, couchgrass, ryegrass, mould or Phadiatop IgE ≥0.35 kU/L at any age, or mould SPT (*Alternaria* or *Aspergillus* spp.)>2 or 3 mm for age ≤2 or>2, respectively.

DOI: https://doi.org/10.7554/eLife.35856.008

The following figure supplement is available for figure 3:

**Figure supplement 1.** Relationship of clusters to food sensitisation, eczema and wheeze.

DOI: https://doi.org/10.7554/eLife.35856.009

later wheeze (*Table 4*). Furthermore, the protected status of CAS2 relative to CAS3 was unlikely to be driven by IgG4, given that CAS3 had greater quantities of both IgE and IgG4.

Although they were highly correlated, IgE, IgG, Th2 cytokine and SPT responses did not overlap perfectly. CAS3 was enriched for individuals with strong signals in all modalities, but there remained individuals within CAS3 and the rest of the cohort who were only responsive in some modalities and not others. Notably, the general direction of IgE, IgG4, SPT and Th2 cytokine signals did not always coincide (*Figure 4—figure supplement 2*).

## Comparison of clusters to existing criteria for atopy

The npEM-derived CAS clusters were partially consistent with traditional atopy thresholds (i.e. any specific IgE ≥0.35 kU/L or SPT ≥ 2 mm at age 2). When we compared CAS clusters with supervised groups created using traditional thresholds (*Supplementary file 1* – table supplement 5), we found that CAS1 most closely matched a non-atopic phenotype (58 of 84 had no specific IgE greater than 0.35 kU/L by age 2). Conversely, CAS2 and CAS3 partially matched traditional criteria for atopy,

**Table 2.** Comparison of HDM-associated immunological variables in CAS clusters

| Variable | Age | Cas1 (N = 88) | Cas2 (N = 107) | Cas3 (N = 22) | P-value (unadjusted) | | | | Feature? |
|---|---|---|---|---|---|---|---|---|---|
| | | Geom. mean (95% CI) | Geom. mean (95% CI) | Geom. mean (95% CI) | Overall | Cas1 vs. 2 | Cas1 vs. 3 | Cas2 vs. 3 | |
| *Total antibody* | | | | | | | | | |
| IgE (kU/L) | 6m | 1.2 (0.69–2) | 2.2 (1.4–3.6) | 21 (12–35) | 1.2E-07 | 0.044 | 6.7E-08 | 2.2E-06 | Yes |
| | 1 | 0.6 (0.29–1.3) | 2 (1.1–3.7) | 43 (17–109) | 2.0E-09 | 0.019 | 4.3E-09 | 5.3E-08 | Yes |
| | 2 | 6.6 (3.5–12) | 17 (12–25) | 187 (131–267) | 1.2E-11 | 0.044 | 4.2E-11 | 1.4E-10 | Yes |
| | 5 | 35 (23–55) | 60 (46–80) | 451 (278–731) | 2.2E-08 | *0.096* | 1.9E-08 | 1.5E-07 | No |
| | 10 | 85 (46–154) | 150 (103–217) | 800 (405–1.6E + 03) | 1.4E-04 | 0.11 | 1.3E-04 | 2.8E-04 | No |
| *HDM antibody* | | | | | | | | | |
| IgE (kU/L) | 6m | 0.018 (0.016–0.02) | 0.019 (0.016–0.022) | 0.033 (0.019–0.059) | 1.9E-03 | 0.47 | 7.9E-04 | 4.2E-03 | Yes |
| | 1 | 0.019 (0.017–0.023) | 0.019 (0.016–0.022) | 0.26 (0.075–0.93) | 1.3E-09 | 0.47 | 2.5E-07 | 4.5E-09 | Yes |
| | 2 | 0.024 (0.019–0.031) | 0.042 (0.029–0.06) | 7.1 (2.7–19) | 2.6E-16 | *0.078* | 2.5E-15 | 3.5E-13 | Yes |
| | 5 | 0.072 (0.041–0.13) | 0.23 (0.12–0.45) | 31 (7.8–127) | 4.2E-09 | 0.015 | 3.8E-09 | 5.1E-07 | No |
| | 10 | 0.37 (0.17–0.8) | 1.3 (0.51–3.4) | 52 (19–144) | 2.9E-06 | *0.068* | 5.7E-07 | 9.7E-05 | No |
| IgG (mg/L) | 1 | 0.21 (0.2–0.23) | 0.23 (0.21–0.25) | 0.29 (0.21–0.39) | 0.042 | 0.34 | 0.012 | *0.07* | Yes |
| | 2 | 0.32 (0.27–0.37) | 0.49 (0.41–0.59) | 0.89 (0.57–1.4) | 1.9E-06 | 2.1E-04 | 3.8E-06 | 7.0E-03 | Yes |
| | 5 | 0.55 (0.42–0.7) | 0.59 (0.46–0.74) | 1.7 (0.88–3.3) | 1.5E-03 | 0.67 | 6.4E-04 | 9.0E-04 | No |
| | 10 | 1.6 (1.3–1.9) | 2.1 (1.8–2.5) | 2.8 (1.9–4.2) | 1.0E-02 | 0.023 | 0.011 | 0.18 | No |
| IgG4 (µg/L) | 6m | 1.5E-04 (1.5E-04–1.5E-04) | 1.7E-04 (1.3E-04–2.1E-04) | 4.6E-04 (9.0E-05–2.4E-03) | 4.9E-03 | 0.37 | 5.2E-03 | 0.024 | Yes |
| | 1 | 1.5E-04 (1.5E-04–1.5E-04) | 6.9E-04 (3.2E-04–1.5E-03) | 0.081 (4.6E-03–1.4) | 1.8E-10 | 5.2E-04 | 6.6E-12 | 2.2E-05 | Yes |
| | 2 | 3.4E-04 (1.8E-04–6.6E-04) | 4.8 (1.7–13) | 61 (8.9–419) | 1.8E-25 | 1.5E-22 | 8.6E-18 | 9.8E-05 | Yes |
| | 5 | 2 (0.48–8.1) | 168 (111–256) | 539 (317–917) | 1.1E-15 | 1.3E-12 | 1.0E-08 | 1.9E-04 | No |
| *HDM cytokine response* | | | | | | | | | |
| IL-13 protein (pg/ml) | 0 | 0.22 (0.066–0.73) | 0.22 (0.076–0.63) | 0.085 (0.011–0.66) | 0.68 | 0.76 | 0.41 | 0.45 | No |
| | 6m | 0.064 (0.022–0.18) | 0.06 (0.025–0.14) | 19 (1.4–244) | 4.6E-06 | 0.98 | 1.7E-05 | 4.1E-06 | No |
| | 5 | 0.13 (0.046–0.37) | 0.32 (0.11–0.87) | 12 (1.2–117) | 2.1E-04 | 0.29 | 7.7E-05 | 5.1E-04 | No |
| IL-5 protein (pg/ml) | 0 | 0.043 (0.018–0.11) | 0.026 (0.013–0.052) | 0.018 (5.0E-03–0.068) | 0.44 | 0.36 | 0.29 | 0.57 | No |
| | 6m | 0.018 (9.2E-03–0.034) | 0.013 (8.9E-03–0.02) | 0.21 (0.012–3.7) | 7.9E-04 | 0.4 | 8.1E-03 | 3.5E-04 | No |

*Table 2 continued on next page*

*Table 2 continued*

| Variable | Age | Cas1 (N = 88) | Cas2 (N = 107) | Cas3 (N = 22) | P-value (unadjusted) | | | | Feature? |
|---|---|---|---|---|---|---|---|---|---|
| | | Geom. mean (95% CI) | Geom. mean (95% CI) | Geom. mean (95% CI) | Overall | Cas1 vs. 2 | Cas1 vs. 3 | Cas2 vs. 3 | |
| | 5 | 0.028 (0.014–0.057) | 0.042 (0.02–0.087) | 2.3 (0.25–22) | **3.2E-06** | 0.45 | **5.7E-06** | **2.0E-05** | No |
| IL-13 mRNA^ | 0 | 1.7E-03 (1.1E-04–0.026) | 6.0E-03 (4.8E-04–0.075) | 6.7E-03 (3.3E-05–1.4) | 0.85 | 0.6 | 0.68 | 0.94 | No |
| | 6m | 1.0E-04 (8.8E-06–1.1E-03) | 3.2E-04 (3.8E-05–2.6E-03) | 2 (0.015–266) | **3.2E-04** | 0.5 | **1.7E-04** | **3.8E-04** | No |
| | 5 | 0.036 (1.6E-03–0.8) | 0.11 (8.8E-03–1.4) | 2.9E + 03 (742–1.1E + 04) | **6.8E-05** | 0.59 | **9.9E-05** | **2.5E-05** | No |
| IL-4 mRNA^ | 0 | 1.4E-06 (6.9E-07–3.0E-06) | 1.9E-06 (7.8E-07–4.4E-06) | 1.0E-06 (1.0E-06–1.0E-06) | 0.71 | 0.65 | 0.6 | 0.47 | No |
| | 6m | 4.6E-06 (1.0E-06–2.1E-05) | 5.1E-06 (1.4E-06–1.8E-05) | 0.54 (6.5E-03–44) | **6.2E-09** | 0.94 | **4.7E-07** | **1.0E-07** | No |
| | 5 | 2.3E-04 (1.7E-05–3.0E-03) | 4.7E-04 (5.3E-05–4.3E-03) | 5.3 (0.082–345) | **4.9E-04** | 0.72 | **4.5E-04** | **3.2E-04** | No |
| IL-5 mRNA^ | 0 | 2.5E-04 (2.1E-05–2.9E-03) | 2.6E-04 (2.8E-05–2.5E-03) | 1.2E-05 (3.1E-07–4.6E-04) | 0.47 | 0.96 | 0.24 | 0.25 | No |
| | 6m | 5.2E-05 (5.6E-06–4.8E-04) | 3.1E-05 (5.2E-06–1.8E-04) | 0.33 (1.3E-03–83) | **1.5E-04** | 0.85 | **2.3E-04** | **1.1E-04** | No |
| | 5 | 0.021 (9.9E-04–0.43) | 0.07 (5.7E-03–0.85) | 246 (7–8.7E + 03) | **1.3E-04** | 0.49 | **7.1E-05** | **1.1E-04** | No |
| | | Prop. (95% CI) | Prop. (95% CI) | Prop. (95% CI) | Overall | Cas1 vs. 2 | Cas1 vs. 3 | Cas2 vs. 3 | |
| *HDM SPT past atopy threshold* | | | | | | | | | |
| Wheal ≥ 2 mm | 6m | 2.3% (0–5.4%) | 1.9% (0–4.5%) | 14% (0–29%) | **0.043** | 1 | *0.054* | **0.035** | No* |
| | 2 | 10% (3.8–17%) | 15% (8.1–22%) | 86% (71–100%) | **2.9E-12** | 0.39 | **8.2E-12** | **1.5E-10** | No* |
| Wheal ≥ 3 mm | 5 | 13% (5.2–20%) | 28% (18–37%) | 81% (63–99%) | **1.5E-08** | **0.022** | **4.6E-09** | **1.0E-05** | No |
| | 10 | 36% (23–49%) | 51% (38–63%) | 78% (57–99%) | **7.4E-03** | 0.11 | **2.7E-03** | *0.06* | No |

Feature?=whether variable was used as a clustering feature or not; geom. mean = geometric mean; PBMC = peripheral blood mononuclear cells; prop. = proportion; SPT = skin prick or sensitisation test. For categorical variables, associations were tested using Fisher exact test; for continuous variables, Kruskal-Wallis and Mann-Whitney-Wilcoxon. Bold text indicates statistical significance (p<0.05); italics indicate near-significance (p<0.10). ^PBMC cytokine responses to HDM above unstimulated control; birth samples (age 0) taken from cord blood (CBMC). *Not used as clustering features, as these are derived variables; the variables from which they were derived (HDM IgE and IgG4) were used instead.

DOI: https://doi.org/10.7554/eLife.35856.014

with CAS3 being an extreme phenotype (all 22 children in CAS3 had some specific IgE ≥0.35 kU/L by age 2).

However, the CAS clusters outperformed IgE/SPT-defined atopy in terms of predicting for age-5 wheeze (likelihood ratio test for clusters vs. IgE/SPT, Chi-squared = 23, p=2.0 × 10^{-6}). In addition, at age 2, 68% of CAS3 were 'sensitised' (any specific IgE ≥0.35 kU/L) to two or more allergens, compared to only 1% and 6% for CAS1 and CAS2 respectively. This emphasised CAS3 as an early- and multi-sensitised phenotype. Finally, fewer members of CAS1 and CAS2 who were IgE- or SPT-responsive prior to age 5 maintained atopic wheeze at age 5 (23% or 24 of 103), compared to CAS3 (76% or 16 of 21). Therefore, the association of IgE and SPT with disease risk varied across clusters. This suggests that fixed atopy thresholds are not sufficient to delineate risk profiles – instead, an unsupervised clustering approach may be more informative.

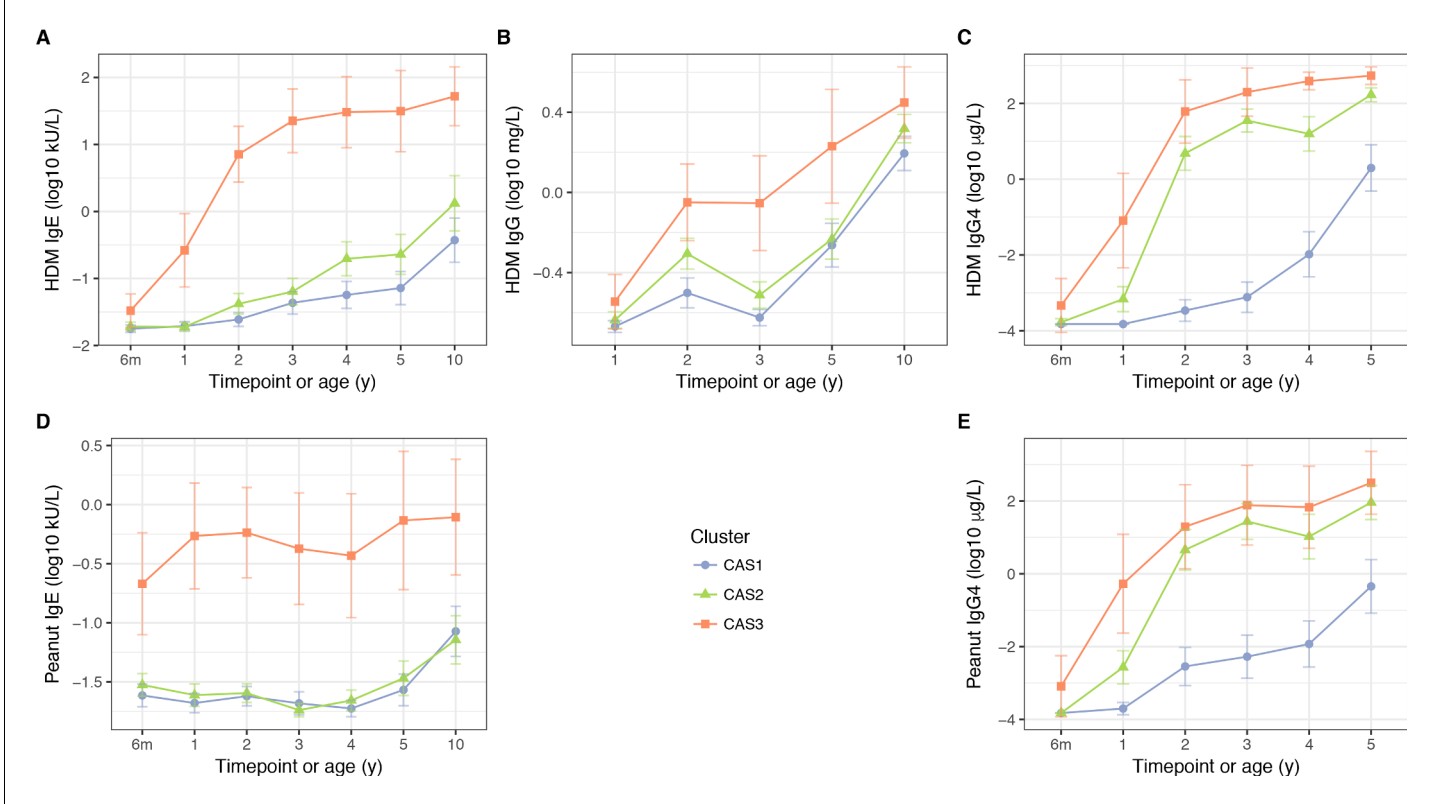

**Figure 4.** HDM IgE (**A**), IgG (**B**) and IgG4 (**C**); and peanut IgE (**D**) and IgG4 (**E**) stratified by cluster and time, in the CAS dataset Points indicate means; bars indicate 95% CI (t-distribution).

DOI: https://doi.org/10.7554/eLife.35856.010

The following figure supplements are available for figure 4:

**Figure supplement 1.** Correlation patterns between IgE vs IgG4 (**A**) and IgE vs IgG (**B**) at age five *p<0.05 for Spearman correlation with Holm correction for multiple testing.

DOI: https://doi.org/10.7554/eLife.35856.011

**Figure supplement 2.** Distinct biological signals of HDM IgE, IgG4, SPT, and Th2 cytokine (IL-13).

DOI: https://doi.org/10.7554/eLife.35856.012

## Comparison of clusters to time-dependent wheeze phenotypes and atopic disease

We mapped the npEM-derived clusters to pre-defined wheezing phenotypes (*Figure 3C*): no wheeze (in the first 3 years of life, or at age 5), transient wheeze (only in first 3 years), late wheeze (only at age 5), and persistent wheeze (both first 3 years and age 5). We found that CAS3 was enriched for persistent wheeze, while individuals in CAS1 or CAS2 tended to have transient or no wheeze. There were rarely any members of CAS with late wheeze (approximately 10%).

In addition to persistent wheeze, CAS3 was also enriched for persistent food sensitisation (peanut IgE ≥0.35 kU/L, or positive egg white or cow's milk SPTs) and persistent eczema: 44% of CAS3 experienced all three (*Figure 3—figure supplement 1*). Almost all individuals in CAS3 had both eczema and food sensitisation from age 6 m onwards, with rates of food sensitisation and wheeze increasing with time (*Figure 3D*). In contrast, CAS1 and CAS2 had low rates of food sensitisation, and declining rates of both eczema and wheeze. These trends lend credence to recent suggestions that the 'atopic march' phenotype (*Bantz et al., 2014*; *Han et al., 2017*) may only be present in a minority of the population (e.g. CAS3) (*Belgrave et al., 2014*).

**Table 3.** Comparison of selected respiratory-disease-related variables in CAS clusters

| Variable | Age (y) | Cas1 (N = 88) | Cas2 (N = 107) | Cas3 (N = 22) | P-value (unadjusted) | | | | Feature? |
|---|---|---|---|---|---|---|---|---|---|
| | | Mean (95% CI) | Mean (95% CI) | Mean (95% CI) | Overall | Cas1 vs. 2 | Cas1 vs. 3 | Cas2 vs. 3 | |
| URI (events per y) | 1 | 2.9 (2.4–3.3) | 2.6 (2.2–3) | 2.5 (1.7–3.3) | 0.59 | 0.34 | 0.5 | 0.96 | Yes |
| | 2 | 3.2 (2.6–3.7) | 2.6 (2.2–3) | 2.5 (1.2–3.8) | 0.19 | 0.19 | 0.12 | 0.34 | Yes |
| | 3 | 2.7 (2.2–3.2) | 2.8 (2.4–3.3) | 2.2 (1.3–3.2) | 0.45 | 0.41 | 0.59 | 0.24 | Yes |
| | 4 | 2.1 (1.7–2.6) | 2.2 (1.8–2.7) | 1.7 (0.77–2.7) | 0.5 | 0.94 | 0.26 | 0.27 | No |
| | 5 | 1.6 (1.1–2) | 1.5 (1.2–1.9) | 0.67 (0.2–1.1) | *0.081* | 0.76 | **0.047** | **0.026** | No |
| LRI (events per y) | 1 | 1.6 (1.2–1.9) | 0.98 (0.76–1.2) | 2 (1.3–2.6) | **4.0E-03** | **0.021** | 0.17 | **2.6E-03** | Yes |
| | 2 | 1.4 (0.98–1.7) | 1 (0.81–1.2) | 2.2 (1.6–2.9) | **2.5E-03** | 0.83 | **6.1E-03** | **2.0E-04** | Yes |
| | 3 | 1 (0.76–1.3) | 0.6 (0.4–0.8) | 1.8 (1.1–2.6) | **6.1E-04** | **0.02** | **0.039** | **2.7E-04** | Yes |
| | 4 | 0.87 (0.52–1.2) | 0.46 (0.3–0.63) | 2 (1.1–2.8) | **1.7E-05** | 0.3 | **3.5E-04** | **1.6E-06** | No |
| | 5 | 0.42 (0.24–0.6) | 0.36 (0.24–0.48) | 0.86 (0.44–1.3) | **0.019** | 1 | **0.011** | **7.5E-03** | No |
| Wheezy LRI (wLRI, events per y) | 1 | 0.47 (0.3–0.63) | 0.24 (0.15–0.34) | 0.64 (0.19–1.1) | *0.054* | **0.036** | 0.61 | *0.065* | Yes |
| | 2 | 0.68 (0.45–0.91) | 0.41 (0.26–0.56) | 1 (0.56–1.5) | **5.2E-03** | *0.063* | *0.066* | **1.7E-03** | Yes |
| | 3 | 0.59 (0.37–0.81) | 0.3 (0.17–0.44) | 1.4 (0.78–2.1) | **4.6E-05** | *0.065* | **2.5E-03** | **6.6E-06** | Yes |
| | 4 | 0.52 (0.25–0.79) | 0.32 (0.18–0.46) | 1.9 (0.95–2.8) | **4.5E-08** | 0.86 | **9.3E-07** | **3.3E-08** | No |
| | 5 | 0.28 (0.13–0.42) | 0.23 (0.13–0.33) | 0.76 (0.36–1.2) | **2.3E-03** | 0.99 | **2.0E-03** | **1.2E-03** | No |
| Febrile LRI (fLRI, events per y) | 1 | 0.36 (0.22–0.51) | 0.28 (0.16–0.4) | 0.55 (0.28–0.81) | **0.025** | 0.24 | *0.071* | **6.4E-03** | Yes |
| | 2 | 0.36 (0.23–0.5) | 0.33 (0.22–0.43) | 0.95 (0.46–1.4) | **0.01** | 1 | **6.1E-03** | **3.8E-03** | Yes |
| | 3 | 0.38 (0.21–0.55) | 0.16 (0.09–0.23) | 0.52 (0.13–0.92) | *0.06* | *0.063* | 0.44 | **0.04** | Yes |
| | 4 | 0.3 (0.13–0.47) | 0.15 (0.064–0.24) | 0.43 (0.16–0.7) | **0.021** | 0.18 | *0.091* | **4.9E-03** | No |
| | 5 | 0.19 (0.082–0.3) | 0.14 (0.06–0.21) | 0.19 (0–0.42) | 0.83 | 0.55 | 0.91 | 0.8 | No |
| | | Prop. (95% CI) | Prop. (95% CI) | Prop. (95% CI) | Overall | Cas1 vs. 2 | Cas1 vs. 3 | Cas2 vs. 3 | |
| >20% *Streptococcus* in first infection-naive NPA sample | 7w | 11% (0.34–23%) | 15% (3.3–26%) | 44% (3.9–85%) | *0.081* | 0.75 | **0.042** | *0.065* | No |
| | 6m | 7.6% (1.6–14%) | 18% (10–26%) | 14% (0–31%) | 0.12 | **0.045** | 0.39 | 1 | No |
| % Healthy NPAs with infection-associated MPGs | 0–2 | 49% (38–59%) | 32% (24–39%) | 62% (47–76%) | **1.2E-03** | **0.013** | 0.2 | **5.5E-04** | No |
| | 2–4 | 46% (37–55%) | 44% (37–51%) | 45% (29–61%) | 0.9 | 0.67 | 0.92 | 0.8 | No |

Feature?=whether variable was used as a clustering feature or not; geom. mean = geometric mean; ARI = acute respiratory infection (lower or upper); LRI = lower respiratory infection; MPG = microbiome profile group; NPA = nasopharyngeal aspirate; prop. = proportion; URI = upper respiratory infection; 7w = 7 weeks. For categorical variables, associations were tested using Fisher exact test; for continuous variables, Kruskal-Wallis and Mann-Whitney-Wilcoxon. Bold text indicates statistical significance (p<0.05); italics indicate near-significance (p<0.10). *Not used as clustering features, as these were derived variables; the variables from which they were derived (URI, LRI, wLRI, fLRI) were used instead.

DOI: https://doi.org/10.7554/eLife.35856.020

**Table 4.** Analysis of selected predictors for age-5 wheeze within each CAS cluster, with demographic covariates (sex, BMI, parental history of asthma)

| Selected predictors for age-5 wheeze | | Cas1 (N = 88) Or (95% CI) | P-value | Cas2 (N = 107) Or (95% CI) | P-value | Cas3 (N = 22) Or (95% CI) | P-value | Or (95% CI) | P-value |
|---|---|---|---|---|---|---|---|---|---|
| LRI (events per y) | 1 | 0.97 (0.71–1.3) | 0.84 | 1 (0.61–1.5) | 0.99 | 0.48 (0.13–1.1) | 0.16 | 1 (0.81–1.2) | 0.92 |
| | 2 | 1.2 (0.88–1.6) | 0.26 | 1.5 (0.97–2.5) | *0.069* | 0.99 (0.34–2.6) | 0.98 | 1.4 (1.1–1.7) | **5.3E-03** |
| | 3 | 2 (1.3–3.2) | **2.3E-03** | 2.6 (1.5–5.3) | **2.7E-03** | 0.98 (0.4–2.6) | 0.96 | 2 (1.5–2.7) | **3.8E-06** |
| | 4 | 2 (1.4–3.4) | **2.0E-03** | 3.6 (1.8–8.3) | **6.5E-04** | 1.9 (0.57–8.4) | 0.32 | 2.5 (1.8–3.6) | **1.5E-07** |
| Wheezy LRI (events per y) | 1 | 1.3 (0.68–2.4) | 0.43 | 1.1 (0.35–3) | 0.83 | 2.6 (0.62–58) | 0.34 | 1.5 (0.98–2.3) | *0.06* |
| | 2 | 1.2 (0.8–2) | 0.33 | 1.6 (0.89–2.9) | 0.12 | 2.4 (0.67–16) | 0.24 | 1.6 (1.2–2.2) | **5.6E-03** |
| | 3 | 2.8 (1.6–5.6) | **1.3E-03** | 3 (1.4–8) | **0.016** | 1.2 (0.43–4.6) | 0.76 | 2.7 (1.8–4.2) | **4.1E-06** |
| | 4 | 2.5 (1.5–5) | **4.0E-03** | 6.3 (2.5–21) | **6.8E-04** | 7.1 (1.2–169) | 0.1 | 3.9 (2.5–6.7) | **5.4E-08** |
| Febrile LRI (events per y) | 1 | 1.6 (0.77–3.6) | 0.21 | 0.84 (0.28–1.9) | 0.71 | 7.3 (0.78–178) | 0.12 | 1.5 (0.93–2.4) | *0.098* |
| | 2 | 1 (0.44–2.2) | 1 | 4.8 (1.8–15) | **3.9E-03** | 1.6 (0.48–10) | 0.5 | 2.3 (1.4–3.9) | **1.2E-03** |
| | 3 | 2 (1–4.8) | *0.08* | 4.3 (1.2–15) | **0.02** | 4.2 (0.55–519) | 0.37 | 2.4 (1.4–4.3) | **2.3E-03** |
| | 4 | 1.8 (0.97–4.1) | *0.092* | 2.6 (0.88–8.3) | *0.082* | 1.1 (0.11–18) | 0.93 | 2.2 (1.3–4) | **5.9E-03** |
| Quartile of % healthy NPAs with infection-associated MPGs | 0–2 | 1 (0.54–1.8) | 0.98 | 1.3 (0.72–2.4) | 0.36 | NA | NA | 1.3 (0.89–1.8) | 0.19 |
| | 2–4 | 0.45 (0.19–0.88) | **0.035** | 1 (0.51–2.1) | 0.9 | NA | NA | 0.8 (0.53–1.2) | 0.24 |
| HDM IgE (kU/L)* | 6m | 8 (0.85–94) | *0.074* | 0.93 (0.14–3.6) | 0.92 | 3.4 (0.26–180) | 0.4 | 2.3 (0.99–5.8) | *0.054* |
| | 1 | 1.5 (0.22–7.8) | 0.65 | 0.54 (0.039–2.3) | 0.51 | 39 (2.5–22000) | *0.082* | 2.7 (1.5–5) | **0.00089** |
| | 2 | 0.93 (0.28–2.5) | 0.89 | 2 (1.2–3.7) | **0.016** | 1.4 (0.38–4.8) | 0.62 | 2 (1.5–2.8) | **2.80E-05** |
| | 3 | 1.4 (0.68–2.9) | 0.32 | 1.5 (0.9–2.4) | 0.12 | 1.5 (0.4–5.2) | 0.55 | 1.7 (1.3–2.2) | **1.00E-04** |
| | 4 | 1.9 (0.94–4.1) | *0.086* | 1.9 (1.2–3.1) | **0.011** | 1.4 (0.31–5.5) | 0.64 | 1.9 (1.5–2.5) | **3.70E-06** |
| HDM IgG4 (μg/L)* | 6m | NA (NA-NA) | 0.55 | 0.053 (NA-6.5e + 24) | 0.99 | 28 (1.7e-34-NA) | 0.99 | 1.4 (0.88–2.6) | 0.17 |
| | 1 | NA (NA-NA) | 0.61 | 1.1 (0.8–1.5) | 0.5 | 0.9 (0.58–1.3) | 0.6 | 1.2 (1–1.4) | *0.053* |
| | 2 | 1.1 (0.71–1.6) | 0.67 | 1.1 (0.85–1.4) | 0.61 | 0.4 (0.038–1.2) | 0.26 | 1.1 (1–1.3) | *0.056* |
| | 3 | 1.1 (0.85–1.5) | 0.35 | 1.1 (0.77–2) | 0.64 | 0.94 (0.19–2.3) | 0.9 | 1.1 (0.98–1.2) | 0.1 |
| | 4 | 1.2 (0.98–1.5) | *0.082* | 0.89 (0.7–1.1) | 0.33 | 0.46 (0.031–5.4) | 0.53 | 1.1 (1–1.3) | **0.034** |
| HDM IgG (mg/L)* | 1 | 25 (0.32–1.6E + 04) | 0.19 | 3.3 (0.16–46) | 0.38 | 5.6E-03 (8.4E-06–0.57) | *0.058* | 2 (0.31–11) | 0.44 |
| | 2 | 0.8 (0.15–3.5) | 0.78 | 0.97 (0.24–3.7) | 0.96 | 0.79 (0.031–18) | 0.88 | 1.3 (0.6–2.9) | 0.48 |
| | 3 | 2.3 (0.14–35) | 0.54 | 0.48 (0.057–2.5) | 0.43 | 3.9 (0.26–96) | 0.34 | 2.1 (0.89–5) | *0.089* |

BMI = body mass index; HDM = house dust mite; LRI = lower respiratory infection. Association analyses performed via generalised linear models (GLM) with demographic covariates: age-5 wheeze ~predictor + sex (male) +BMI at age 3 + paternal history of asthma +maternal history of asthma. Bold text indicates statistical significance (p<0.05); italics indicate near-significance (p<0.10). *Odds ratio (OR) is for every 10-fold increase in IgE, IgG4 or IgG.
DOI: https://doi.org/10.7554/eLife.35856.021

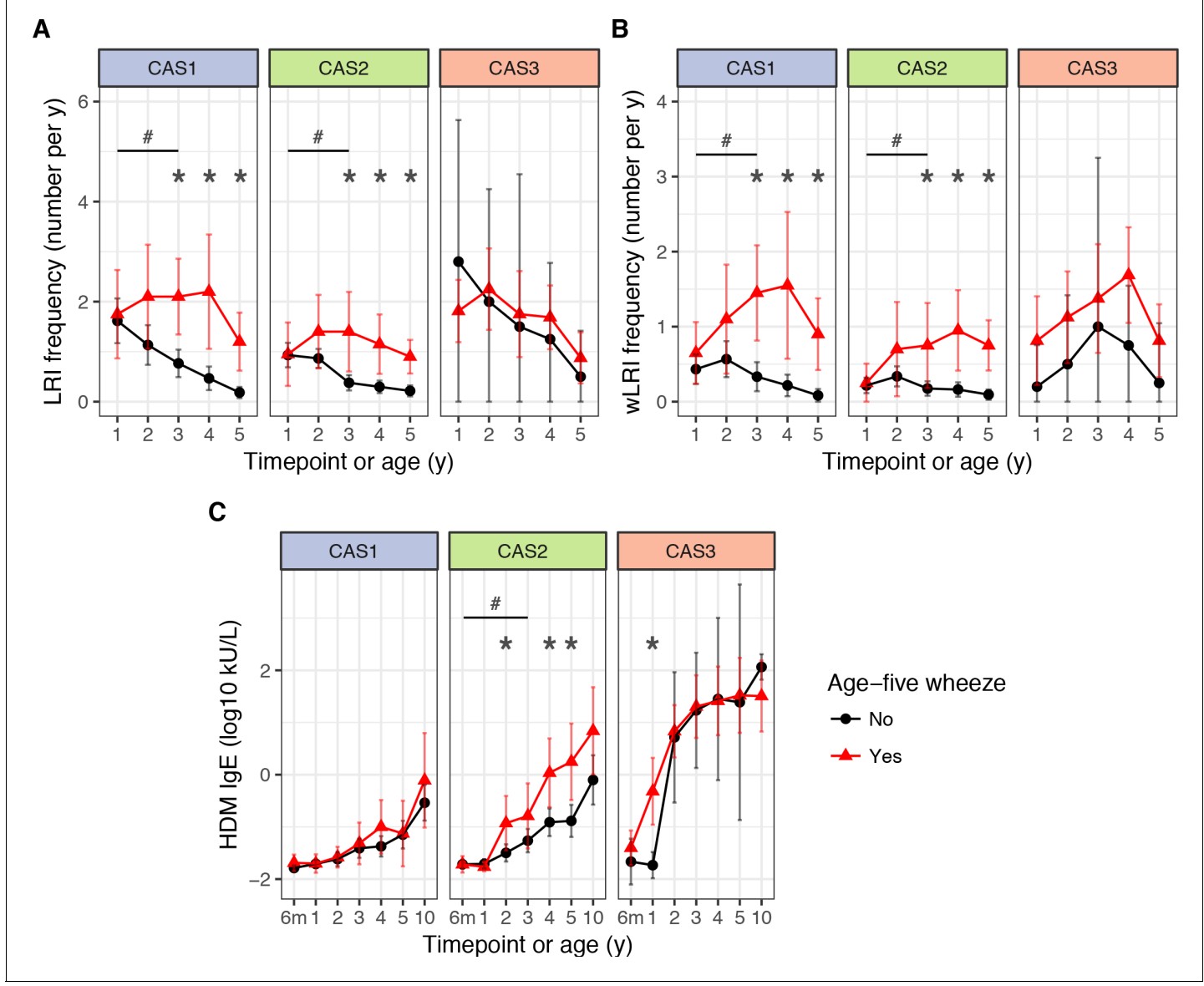

**Figure 5.** LRI frequency (A), wheezy LRI (wLRI) frequency (B), and HDM IgE (C), stratified by age-5 wheeze status, cluster and time, in the CAS dataset. Points indicate means; bars indicate 95% CI (t-distribution). #$p<0.05$ for repeated-measures ANOVA across timepoints from the first 3 years of life (see *Table 4*). *$p<0.05$ for Mann-Whitney-Wilcoxon comparison within each timepoint.

DOI: https://doi.org/10.7554/eLife.35856.015

The following figure supplements are available for figure 5:

**Figure supplement 1.** A 'simple' decision tree generated by recursive partitioning from CAS data, with breakdown of tree clusters by actual CAS npEM-derived clusters.

DOI: https://doi.org/10.7554/eLife.35856.016

**Figure supplement 2.** Decision tree generated by recursive partitioning from CAS data, excluding Phadiatop assay variables.

DOI: https://doi.org/10.7554/eLife.35856.017

**Figure supplement 3.** A 'comprehensive' decision tree generated by recursive partitioning from CAS data, given CAS npEM-derived clusters and age-5 wheezing status.

DOI: https://doi.org/10.7554/eLife.35856.018

**Figure supplement 4.** Comparison of predictors for age-5 wheeze in CAS and COAST clusters.

DOI: https://doi.org/10.7554/eLife.35856.019

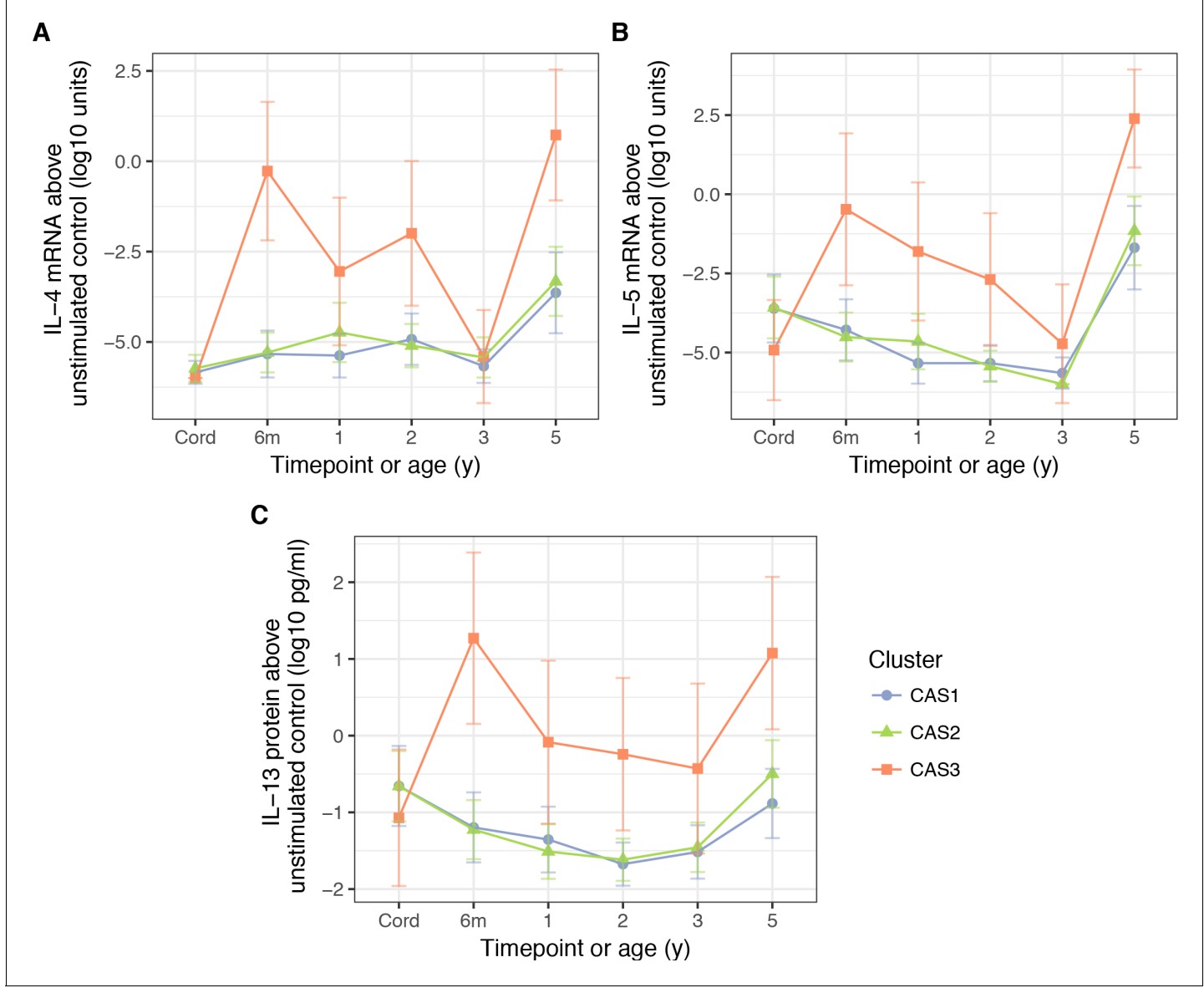

**Figure 6.** PBMC expression of IL-5. (A) and IL-4 mRNA (B), as well as IL-13 protein (C), in response to stimulation HDM, stratified by cluster and time (CAS) Cord = cord blood sample collected at birth.Points indicate means; bars indicate 95% CI (t-distribution).
DOI: https://doi.org/10.7554/eLife.35856.022

## Relationship with the nasopharyngeal microbiome

Previous studies suggest an association between asthma risk and early-life disruption of the respiratory microbiome, especially colonisation with *Streptococcus* spp. in the first 7 weeks of life (*Teo et al., 2015*). In this study, using the same data and definitions, we found that CAS3 was overrepresented by individuals who had >20% relative abundance of *Streptococcus* in their first infection-naive healthy NPA, within the first 7 weeks of life (44% versus 11% and 15% in CAS1 and CAS2, respectively; Fisher exact test, unadjusted p=0.042 and 0.065, respectively; *Table 3*).

Furthermore, Teo et al and others (*Teo et al., 2015*; *Bisgaard et al., 2007*) previously found that transient incursions with certain MPGs (*Streptococcus, Haemophilus, Moraxella* spp.) were associated with increased frequency and severity of subsequent LRIs and wheezing disease. Here, we found that proportion of these infection-associated MPGs in healthy samples from age 0 to 2 was greater in CAS3 (62% vs. 49% and 32% in CAS1 and CAS2, respectively; Fisher exact test,

unadjusted p=0.2 and $5.5 \times 10^{-4}$, respectively; *Table 3*). This finding was independent of LRI and wLRI frequency (GLM; p<0.05 for model predicting group membership, with age-2 LRI and wLRI as covariates). On the contrary, there were no associations between cluster membership and health-associated MPGs (*Corynebacterium*, *Alloiococcus*, *Staphylococcus* spp.; *Supplementary file 1* – table supplement 3E).

Recent work by *Teo et al., 2017*) suggested that infection-associated MPGs in early life were predictive for age-5 wheeze in atopic children, while in non-atopic children they were predictive for transient wheeze. In this study, with the same cohort, we noted a similar trend for infection-associated MPGs from age 0 to 2, in relation to transient wheeze in 'non-atopic' CAS1 (GLM, OR 3.6 per percent, p=0.17, with demographic covariates). Surprisingly, there was evidence that infection-associated MPGs in later samples (from age 2 to 4) were *protective* against age-5 wheeze in CAS1 (OR 0.086 per percent, 0.45 per quartile, p=0.034 and 0.035, respectively; *Table 4*). Infection- and health-associated MPGs were otherwise not associated with age-5 wheeze within the other clusters.

## External replication of clusters in MAAS and COAST

The trajectories described by the CAS npEM clusters were replicated in two cohorts – the Manchester Asthma and Allergy Study (MAAS) (*N* = 1085) (*NAC Manchester Asthma and Allergy Study Group et al., 2002*) from Manchester, UK, and the Childhood Origins of Asthma Study (COAST) (*N* = 289) from Wisconsin, USA (*Lemanske, 2002*). After applying our npEM classifier to these external cohorts (materials and methods), we found that individuals classified into 'Cluster 3' (MAAS3/COAST3) had a persistent disease phenotype extending into late adolescence, with consistently high rates of parent-reported wheeze and physician-diagnosed asthma from birth to age 16. The other two clusters (Cluster 1 = MAAS1/COAST1; Cluster 2 = MAAS2/COAST2) appeared to be low-risk (*Figure 7A,B,D*).

MAAS3 and COAST3 exhibited stronger IgE expression (total, HDM, cat, dog) from ages 1 to 8 (*Figure 7C,E*), compared to other clusters in each dataset. Like CAS3, COAST3 demonstrated elevated PBMC expression of Th2 cytokine protein (IL-5 and IL-13) in response to HDM stimulation at age 3 (*Figure 7F*). This was not replicated in MAAS3, but previous work in MAAS had identified that a strong PBMC Th2 response (IL-5, IL-13) to HDM stimulation at age 8 was associated with increased risk of HDM sensitisation and asthma (*Wu et al., 2015*). Nonetheless, MAAS3 was overrepresented in 'early-sensitised' and 'multiple sensitised' phenotypes described by *Lazic et al. (2013)* from SPT and IgE data. Approximately 86% of individuals in MAAS3 belonged to either one of these two phenotypes, although only 13% of individuals in these two phenotypes were accounted for by MAAS3.

Furthermore, when we explored potential predictors of wheeze phenotypes and asthma diagnosis in later childhood, we found that the clusters in COAST were very similar to those in CAS. In COAST1, LRI and wLRI frequency at age 2 were predictive of asthma diagnosis at age 6 (GLMs with demographic covariates, p=0.02 and 0.02, respectively), while in COAST2, HDM IgE at age 3, and LRI, wLRI and fLRI frequencies at age were all predictive (GLMs, p<0.05 for all) (*Figure 5—figure supplement 4*). Although the timing and magnitude of associations differed between cohorts, this reaffirmed wheeze in Cluster 1 as being primarily non-atopic in origin, while wheeze in Cluster 2 appeared to be driven by both non-atopic and atopic factors.

We re-applied npEM classification to CAS using only those features present in MAAS or COAST. For MAAS and COAST features, the subsequent clusters bore 79% and 72% concordance with the original CAS clusters, respectively. In both cases, concordance was excellent for Cluster 3 – all 22 members of the original CAS3 were correctly assigned to Cluster three after re-applying npEM. Therefore, CAS3, COAST3 and MAAS3 likely represent very similar phenotypes.

## Internal stability and validity of CAS clusters

We checked the stability and validity of the CAS clusters with leave-one-out (LOO) analysis, Jaccard indices and silhouette widths. The average Jaccard indices from leave-one-individual-out analysis were 0.77, 0.76, and 0.85 for CAS1, 2 and 3, respectively. For leave-one-feature-out analysis, the average indices were 0.65, 0.60, and 0.74, respectively. This demonstrates that the clusters, especially CAS3, were relatively resilient to minor changes in sampling or feature selection.

In relation to internal validity of the CAS clusters, average silhouette widths were universally poor, at 0.05, 0.06 and 0.002 for CAS1, 2, 3, respectively, with an average for all three clusters of 0.05

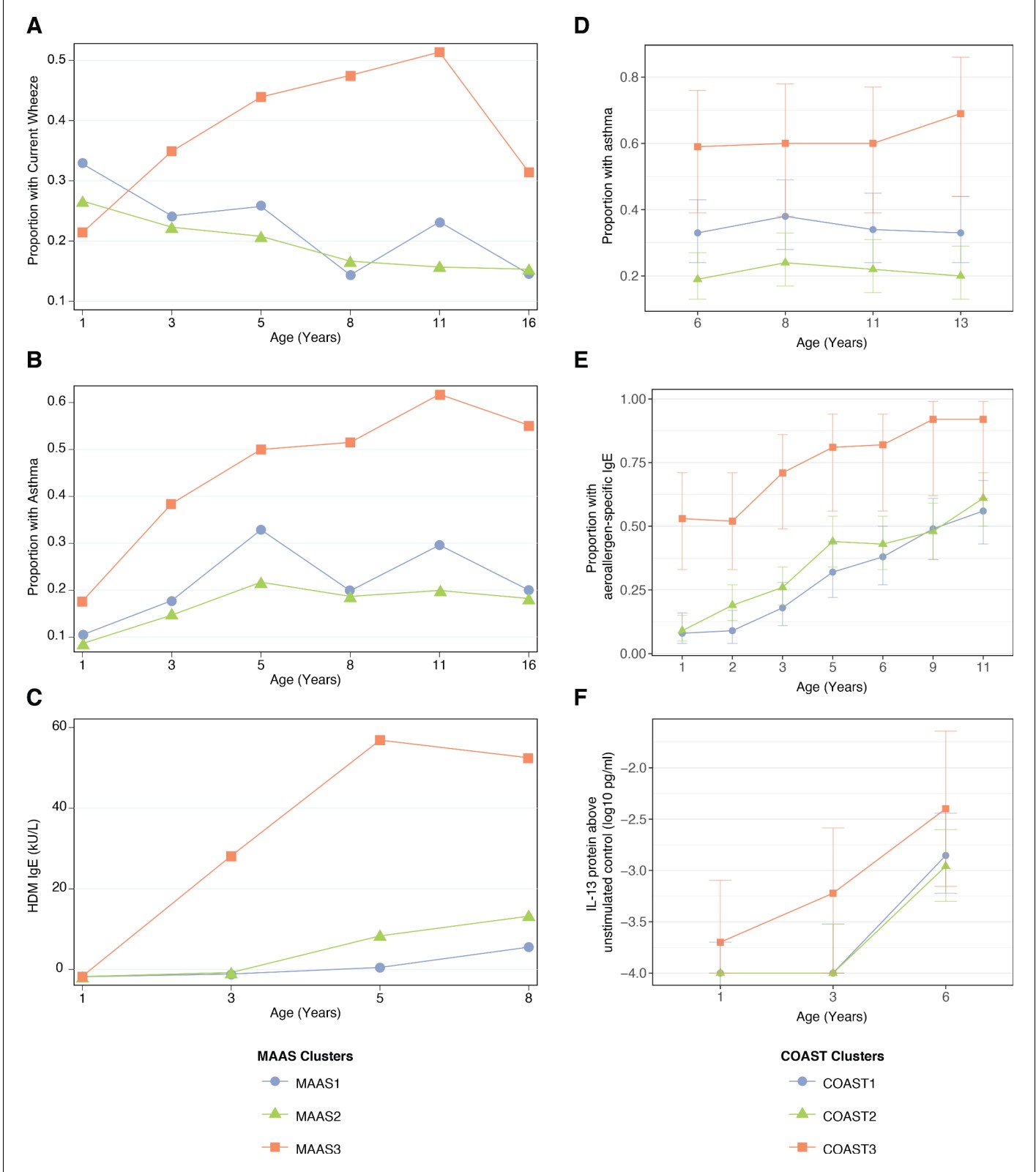

**Figure 7.** Description of npEM-derived clusters in external cohorts: in MAAS, incidence of wheeze. (**A**), asthma diagnosis (**B**), and HDM IgE levels (**C**); in COAST, incidence of asthma diagnosis (**D**), proportion of individuals with detectable aeroallergen-specific IgE levels (**E**), and PBMC protein expression of IL-13 following HDM stimulation above unstimulated control (**F**) MAAS cohort (N = 934) was classified using npEM model from CAS, into MAAS1

*Figure 7 continued on next page*

*Figure 7 continued*

(N = 199, 21%), MAAS2 (N = 692, 74%) and MAAS3 (N = 43, 5%); these correspond to CAS clusters CAS1, 2 and 3, respectively. COAST cohort
(N = 285) was similarly classified into COAST1 (N = 105, 37%), COAST2 (N = 151, 53%) and COAST3 (N = 29, 10%).

DOI: https://doi.org/10.7554/eLife.35856.023

(*Figure 1—figure supplement 2*). Silhouette widths were particularly suboptimal with CAS3, with at least half of those classified having negative values. The overall poor internal validity of the clusters may be due to the large-scale and exploratory nature of our approach – the metric may have been obscured by intra-cluster heterogeneity in other variables that were not particularly important for determining cluster membership. However, it must be noted that all clusters on average yielded positive silhouette widths, and as observed in the rest of the results, they were all relatively homogeneous in terms of the outcomes of interest (wheeze status, allergic disease phenotypes).

## Decision tree analysis

Decision tree analysis on the CAS dataset, using all available predictors from all timepoints, created a 'Simple Tree' with two decision nodes and three end nodes (*Figure 5—figure supplement 1*). This tree had 89% accuracy in retrieving cluster memberships from the original npEM model, where accuracy is calculated as percentage overlap of tree clusters with original CAS clusters. We found that membership in the CAS3-equivalent tree cluster was a better predictor for age-5 wheeze (likelihood ratio test, Chi-squared = 19, $p < 1 \times 10^{-5}$) than traditional thresholds for atopy based on IgE and SPT measurements at age 2. IgG4-related variables best separated CAS1 from other clusters, while IgE-related variables best separated CAS2 and CAS3. Explicitly forcing the exclusion of Phadiatop variables from tree analysis caused these thresholds to be replaced with allergen-specific assays (HDM IgE for Phadiatop IgE, *Figure 5—figure supplement 2*) in a way that is consistent with correlation patterns amongst IgE and IgG4 variables (*Supplementary file 1* – table supplement 6).

We also constructed a 'Comprehensive Tree' that best split individuals into six groups, based on cluster membership crossed with age-5 wheeze status (*Figure 5—figure supplement 3*). We thus identified nodes that were consistent with predictors for wheeze found in the previous regression analyses (*Table 4*), combined with nodes from the Simple Tree (*Figure 5—figure supplement 1*). The Comprehensive Tree had 77% accuracy in recovering both cluster membership and wheeze status. In terms of identifying pure wheeze status at age 5, the accuracy of the tree was 84%, with a positive predictive value (PPV, or precision) of 72%, negative predictive value (NPV) of 88%, sensitivity (recall) of 71% and specificity of 89%. The Comprehensive Tree was more successful in flagging age-5 wheeze (likelihood ratio test, Chi-squared = 60, $p = 6.1 \times 10^{-13}$), compared to the traditional atopy thresholds described previously.

## Discussion

We have used model-based cluster analysis to uncover clusters of children with differential asthma susceptibility. Specifically, there was a high-risk group (Cluster 3) characterised by very early allergen-specific Th2 activity; early sensitisation to multiple allergens including food allergens; and concurrent frequent respiratory infections – resulting in high incidence of atopic persistent wheeze. We also found a lower risk cluster (Cluster 2), with limited or delayed elevation in IgE – this resulted in a lower incidence of mixed (atopic and non-atopic) wheeze. Finally, there was a low-risk cluster (Cluster 1) which exhibited occasional and transient infection-related wheeze, with minimal allergen sensitisation. These clusters were replicated in external datasets, suggesting relevance across populations. Summaries of key findings are given in *Table 5* and *Figure 2*.

## Cluster three is a high-risk, multi-sensitised, atopic phenotype

Cluster 3 represented a multi-sensitive or polysensitised phenotype (*Bousquet et al., 2015*). In CAS3, not only was total IgE elevated, but specific IgE were also raised for most allergens. Three in four CAS3 individuals were sensitised (specific IgE ≥0.35 kU/L) to two or more allergens. In our external replication with MAAS, we observed a large overlap between our predicted high-risk phenotype (MAAS3) and the multiple atopy phenotype from *Lazic et al., 2013*). This was consistent

**Table 5.** Key findings from cluster analysis

Certain childhood populations may be broadly split into three clusters, each representing a unique trajectory of immune function and susceptibility to respiratory infections: low-risk non- atopic Cluster 1 with transient wheeze; low-risk but allergy-susceptible Cluster 2 with mixed wheeze; and strongly-atopic high-risk Cluster 3 with persistent wheeze.

Cluster 3 is consistent with an early-sensitised and multi-sensitised phenotype.

HDM hypersensitivity is an important predictor of wheeze in allergic or allergy -susceptible individuals.

Food and peanut hypersensitivities are important contributors to membership in high-risk Cluster 3. This may be pathophysiologically related to eczema, multi-sensitisation and the atopic march.

In CAS, IgG4 flags for clusters with susceptibility to atopic disease (CAS2 and CAS3), while early and multiple-allergen elevation in IgE predicts frank atopic disease. The pathophysiological role of IgG4 remains unclear.

Allergic and infective processes act additively to intensify airway inflammation during respiratory pathogen clearance. Some (Cluster 3) may be more susceptible to this effect than others that lack strong allergic sensitisation (Cluster 1).

Tests for atopy (IgE, SPT, cytokines) do not overlap perfectly. Therefore, atopy may be better defined by the composite result from a battery of tests encapsulated in a predictive model, rather than just a single test or threshold.

The microbiome acts differently on asthma risk depending on cluster membership. In CAS, early-life asymptomatic colonisation with infection-associated MPGs is associated with risk of persistent wheeze in allergy-susceptible clusters (CAS2, CAS3), while it is potentially protective in non-atopic children (CAS1)

Different childhood populations may share similar trajectories of asthma susceptibility, but there may be subtle differences in terms of the types of tests, allergens, or biological signals that are most informative (SPT, IgE, cytokines, etc.).

DOI: https://doi.org/10.7554/eLife.35856.024

with findings from other studies, where the severely atopic and polysensitised subpopulation was at greater risk of both wheezing disease and reduced lung function (*Hose et al., 2017*).

It is not currently known what is fundamentally producing the strong atopic predisposition in Cluster 3. It is possible that inherited (genetic/epigenetic) or environmental factors (including in utero or perinatal exposures) may be involved, and these should be targets for future investigations. The overrepresentation of males in CAS3 is consistent with the consensus that young boys are at greater risk for asthma than young girls; this was traditionally believed to be due to intrinsic sex differences in airway diameter (*Almqvist et al., 2008*). However, our cluster analysis did not employ any clustering features related to airway size. This suggests that other sex-related factors could be involved, such as differences in immunity and allergic susceptibility. Allergic sensitisation is more frequent amongst prepubescent boys than girls (*Gabet et al., 2016*; *Kim et al., 2014*), and this may be linked to differences in cytokine responsiveness. However, not all boys were clustered into Cluster 3; and sex was not found to be a determinant for either IgE levels or cytokine response in CAS.

We did observe that CAS3 overlapped strongly with both persistent food sensitisation and eczema, and that persistent wheeze co-occurred with early sensitisation and eczema. This suggests that the 'atopic march' may play a role in CAS3. Early disruption of the skin barrier and exposure to certain food allergens may act in concert to promote and entrench the atopic phenotype, through the activation of cytokine pathways involving TSLP, IL33 and IL25 (*Bantz et al., 2014*; *Han et al., 2017*). Although recent research has suggested that very few children actually follow the disease trajectory of the atopic march (*Belgrave et al., 2014*), we hypothesise that it remains relevant to a small but important high-risk subpopulation, who may potentially benefit from early interventions targeted at halting the progression of disease.

## Role of early-life HDM hypersensitivity

In all three cohorts (CAS, MAAS, COAST), house dust mite (HDM) sensitivity was an important determinant of atopic disease risk. HDM was a strong predictor for both CAS3 membership and later childhood wheeze in CAS2, as well as being a 'dominant' allergen in the Phadiatop Infant assays. CAS3 in particular exhibited early and extreme HDM hypersensitivity, with prematurely-elevated HDM IgE, as well as PBMC Th2 response (IL-4, 5, 9, 13) to HDM stimulation. Similar phenomena were seen with MAAS3 and COAST3. The importance of HDM hypersensitivity in driving allergic disease in some populations is well-described in the literature (*Thomas et al., 2010*; *Calderón et al., 2015*). Previous findings from MAAS and a similar cohort RAINE (*Wu et al., 2015*) have shown a confluence of high HDM IgE, as well as PBMC Th2 cytokine levels such as IL-13 and IL-5, in discrete subsets of the population. However, we did observe that in other clusters (CAS1 and CAS2), some

individuals with purported HDM sensitisation (IgE >0.35 kU/L) did not produce detectable Th2 responses; the reverse was also true, where Th2 response did not necessarily result in high IgE. It may be the case that there is high intra-individual variation in IgE and cytokine responses, or stochastic variation in detectability of IgE or cytokine, which may obscure association analyses. Regardless, early and strong Th2 cytokine responses against HDM indicate a high-risk phenotype.

### Role of early-life food and peanut sensitisation

Interestingly, early-life peanut IgE was a strong delineator between high-risk CAS3 and lower-risk CAS1 and 2. There is evidence in the literature for transmission of peanut allergen in utero or via breastmilk (*Vadas et al., 2001*; *DesRoches et al., 2010*), as well as early sensitisation via home environmental exposure, especially in those with concurrent eczema or a predisposing filaggrin (*FLG*) mutation that may allow transcutaneous infiltration of allergen (*Brough et al., 2013*; *Brough et al., 2014*). The strong correlation between Phadiatop and peanut IgE in the first year of life suggests that either peanut reactivity is significant at this earlier timepoint, or that 'peanut-specific IgE' is cross-reactive and representative of some other allergen hypersensitivity. The fact that this correlation exists within each cluster (*Supplementary file 1* – table supplement 6) suggests that it is not caused solely by differences between low- and high-risk clusters (CAS1/CAS2 vs. CAS3). There is a possibility that peanut IgE is a marker for a broader phenotype of early and unremitting sensitisation to multiple food allergens (peanut, cow's milk, eggwhite), as we had observed in CAS3. However, it is unlikely that premature exposure to food allergen is the lone driver for sensitisation and disease, given that well-timed oral exposures to common food allergens (e.g. within 4 to 6 months of age) may actually be protective (*Koplin et al., 2010*). There is some evidence that quantity (minute vs. abundant), route (skin vs. oral) and timing (early vs. late) of exposure are key modifiers of risk (*Han et al., 2017*). Ultimately, an underlying atopic predisposition linked to early-life exposure to food allergen may be driving the high-risk phenotype in Cluster 3.

### IgG4 separates individuals susceptible to atopic wheeze from those who are not

In our study, neither IgG nor IgG4 were strong predictors or protectors of wheeze. However, IgG4 was a strong delineator of cluster membership in CAS, with individuals from CAS2 and CAS3 having elevated IgG4 across all specificities compared to CAS1. Vulnerability to early IgE-driven respiratory disease ('atopic wheeze') can be seen in these same individuals –in CAS2 where HDM IgE is predictive for later wheeze, and in CAS3 where both wheeze frequency and IgE are elevated. Hence, although there had previously been doubt about the efficacy of IgG4 as a marker for atopy (*EAACI Task Force et al., 2008*), our study suggests that IgG4 is still relevant for determining atopic risk, especially when used in combination with IgE.

The underlying biology behind the association of IgG4 with susceptibility to 'atopic wheeze' is unclear. Th2-related pathways drive production of both IgE and IgG4, with IgG4 predominating when modified by concurrent IL-10 signalling (*Davies and Sutton, 2015*). In susceptible individuals, IgG4 production likely precedes isotype switching to frank IgE production (*Aalberse, 2011*). Multiple studies have reported that IgG4 is correlated with induced tolerance following desensitisation immunotherapy with high-dose allergen treatment (*Davies and Sutton, 2015*). However, based on this study alone, we cannot observe any protection from naturally elevated IgG4 levels. Our group had previously suggested, using data from another cohort (*Holt et al., 2016*), that IgG and specifically IgG1 may provide endogenous protection against IgE-associated wheeze in children experiencing natural (low-level) exposure to aeroallergen. In this present study, IgG1 was not measured.

### The role of respiratory infection and nasopharyngeal microbiome in childhood wheeze differs across different clusters

The co-occurrence of elevated IgE and LRI frequency in CAS3, as well as their predictive effect in CAS2, are consistent with previous findings from CAS (*Holt et al., 2010*; *Teo et al., 2015*; *Kusel et al., 2007*). They lend support to the theory that allergic and infective processes act additively to intensify airway inflammation during respiratory pathogen clearance, which in turn drives progression towards persistent wheeze (*Holt and Sly, 2012*). In addition, our cluster analysis suggests that the pathologic effect of this interaction may be stratified in discrete subpopulations,

rather than acting in a strictly dose-dependent fashion across the entire cohort. There may be subsets of children (CAS2 and CAS3) who are more susceptible to the effects of this viral-atopy interaction. On the other hand, pathogen clearance in infected non-atopic (CAS1) subjects may be more efficient, due to lack of susceptibility to the pro-inflammatory effects of atopic co-stimuli. This produces lower levels of 'bystander' inflammatory damage to airway tissues, with opportunity for recovery, resulting in a less severe wheeze phenotype.

Of particular note is that, while both CAS1 and CAS2 have LRI and wLRI frequencies as predictors for age-5 wheeze, CAS2 also has fLRI, particularly at age 2. This, along with the general higher incidence of fLRI in CAS3, is consistent with previous findings from CAS (*Holt et al., 2010*; *Teo et al., 2015*). It suggests that symptomatically severe infections, correlating with severe airway inflammation, may be more potent in causing persistence of wheeze, specifically among those who are 'atopic' (CAS2 and CAS3).

In addition, even during periods of good health, the upper respiratory microbiome played a role in determining later childhood wheeze. Its effect interacted with cluster membership, as well as the age at which the microbiome changes occurred. CAS3 was enriched for early-life infection-associated MPGs (*Streptococcus*, *Moraxella*, and *Haemophilus*). This was consistent with the previous finding by *Teo et al., 2017*) that early-life infection-associated MPGs were predictive of age-5 wheeze only within atopic individuals (as defined by IgE alone). Interestingly, in our current study, we found a protective effect of infection-associated MPGs from age 2 to 4 in CAS1. We hypothesise that those without atopy-related immune dysfunction are able to maintain a healthy trajectory by responding appropriately to stimuli from potential pathogens that colonise the respiratory tract, thus achieving protection against future (non-atopic) wheeze. This is akin to the 'hygiene hypothesis': exposure to a greater repertoire of pathogen-derived antigens may facilitate maturation of immune functions against said pathogens. Meanwhile, individuals with a predisposing immune dysfunction (i.e. 'atopy' manifesting in early-life allergic sensitisation) may be responding in a maladaptive manner to these microbes (*Holt and Sly, 2012*). This may result in inability to clear potential pathogenic bacteria, or shaping of aberrant immune responses – with subsequent effects on airway inflammation and wheeze.

## Implications for cluster analysis in asthma research

In this study, we applied mixture modelling to generate clusters from biological data. Similar methods such as latent class analysis (LCA) have previously been used in asthma research – for instance, LCA was applied to SPT and IgE measurements from MAAS to determine different patterns of allergen sensitisation and subsequent disease (*Lazic et al., 2013*). However, LCA is limited to categorical clustering features, so measures of sensitisation in that study were thresholded (e.g. IgE levels were split into <0.35 kU/L, 0.35 to 100 kU/L, and >100 kU/L). The method also assumed that these thresholds have the same relevance across all timepoints; that thresholds applied equally to all allergens; and that all allergens contributed equally to disease susceptibility profiles. Mixture modelling is an extension of LCA in that it does not require categorical variables or predetermined thresholds. Furthermore, non-parametric mixture modelling (npEM) does not require input features to have Gaussian distributions. Previous studies have used mixture models to explore phenotypes in adult asthma based on clinical measurements (*Janssens et al., 2012*; *Newby et al., 2014*; *Burte et al., 2015*), and one of our own studies previously looked at cytokine expression patterns of PBMCs from children in response to HDM stimulation (*Wu et al., 2015*). Our study is the first to apply non-parametric mixture modelling to data representing immune and respiratory health in early childhood, and to investigate possible predictors of disease within each cluster.

Currently, mixture models are limited by an unproven 'track record'; a lack of consensus about best protocols for data processing and analysis; instability or inconsistency of clusters; difficulty in interpretation of results; and uncertainty regarding the validity of certain assumptions that accompany models (*Deliu et al., 2016*). Other methods of cluster analysis have similar problems, and while they have been applied frequently to asthma research, they have also produced a confusing myriad of phenotypes. The nature of cluster phenotypes is highly dependent on the type of features entered into the clustering algorithm. Clustering features that represent final clinical endpoints, such as markers of severity, may produce more heterogeneous clusters, as different pathological trajectories can arrive at similar endpoints. Some cluster phenotypes may contradict with each other, or may not be easily interpreted. Recently, *Schoos et al. (2017)* identified that, unlike our study, asthma

was *not* as strongly associated with prominent HDM or peanut hypersensitivity in a Danish birth cohort (COPSAC) as other patterns of sensitisation (especially cat, dog and horse). However, we note that they used thresholded IgE >0.35 kU/L to build their clusters. Other differences may emerge due to heterogeneity across different populations; geographical differences in environmental exposures and allergen sensitisation; and differences in testing procedures and phenotype definitions at different sites. COPSAC, CAS and COAST were cohorts enriched for high-risk individuals – each child had at least one parent with a history of atopic disease – while MAAS had no such recruitment criterion. Because of variability in findings, there has been wariness and scepticism among clinicians regarding the utility of mixture models and machine learning (*Chen and Asch, 2017*). Ultimately, one may argue that discrepancies in our findings serve as a caution against the blind application of 'algorithms' without due consideration of subtleties in target population and environment.

Nonetheless, what we have demonstrated here is the vast potential of cluster analysis. We have discovered clusters in an unsupervised and exploratory fashion, described them comprehensively, replicated our findings in multiple datasets, and compared our clusters with other existing phenotypes. In doing so, we have generated some new and interesting insights about the nature of atopy and asthma risk. Our results build on previous findings (*Frith et al., 2011*; *Klink et al., 1990*) demonstrating that the concept of atopy, as an intrinsic or heritable predisposition to allergic disease, is more complicated than what could be described by dichotomies or thresholds. We have also demonstrated that addressing subgroup differences via cluster analysis allows for identification of intra-cluster disease predictors. In the future, clusters may be further characterised by other aspects of asthma pathophysiology, such as genomics, transcriptomics, and epigenomics.

### Concluding statements

The results of our study strongly support the future use of predictive models with more precise and subgroup-driven representations of atopy or other relevant pathophysiology. We argue for ongoing collaboration between research groups in terms of refining methodology, answering questions unique to certain populations, and comparing cluster phenotypes arising from different algorithms and datasets. We believe that, as clustering methods become more frequently used, we will gradually develop better consensus on how such methods are best applied to biomedical phenomena. By continuing with these approaches, we can hopefully move away from fixed thresholds to more sophisticated formulations of risk, which will then improve future attempts at targeted screening, prevention and treatment of asthma. These approaches are already being applied to other heterogeneous diseases, and in the future computerised tools may be designed to embody the sum knowledge from these approaches. Such approaches can eventually help clinicians and scientists achieve a fuller understanding of pathophysiology, and hence better predict and manage human disease.

## Materials and methods

**Key resources table**

| Reagent type (species) or resource | Designation | Source or reference | Identifiers |
|---|---|---|---|
| Biological sample (*Homo sapiens*) | Childhood Asthma Study (CAS) | DOI: 10.1016/j.jaci.2005.06.038 | Microbiome sequencing data: NCBI GenBank SRP056779 |
| Biological sample (*Homo sapiens*) | Childhood Origins of Asthma Study (COAST) | PMID:12688623 | NA |
| Biological sample (*Homo sapiens*) | Manchester Asthma and Allergy Study (MAAS) | PMID:12688622 | NA |
| Software, algorithm | The R project for Statistical Computing | ISBN:3-900051-07-0 | RRID:SCR_001905 |
| Software, algorithm | ggplot2 | ISBN:978-3-319-24277-4 | RRID:SCR_014601 |

*Continued on next page*

*Continued*

| Reagent type (species) or resource | Designation | Source or reference | Identifiers |
|---|---|---|---|
| Software, algorithm | mixtools | DOI: 10.18637/jss.v032.i06 | NA |
| Software, algorithm | rpart | *Therneau and Atkinson, 2015*. Package 'rpart'. URL: https://cran.r-project.org/web/packages/rpart/rpart.pdf | NA |
| Software, algorithm | epiDisplay | *Chongsuvivatwong, 2015*. Package 'epiDisplay' URL: https://cran.r-project.org/web/packages/epiDisplay/epiDisplay.pdf | NA |

## Patients and study design in CAS

Our discovery dataset was the Childhood Asthma Study (CAS), a prospective birth cohort ($N$ = 263) operated by the Telethon Kids Institute from Perth, Western Australia (*Kusel et al., 2005*). The goal of CAS was to describe the risk factors and pathogenesis of childhood allergy and asthma. Further details of CAS have been reported previously (*Kusel et al., 2005*; *Hollams et al., 2009*; *Holt et al., 2010*; *Teo et al., 2015*; *Hollams et al., 2017*).

In CAS, expectant parents were recruited from private paediatric clinics in Perth during the period spanning July 1996 to June 1998. Each child who was born and subsequently recruited had at least one parent with physician-diagnosed asthma or atopic disease (hayfever, eczema). The child was then followed from birth till age 10 at the latest, with routine medical examinations, clinical questionnaires, blood sampling at multiple time points (6–7 weeks, 6 months, 1 year, 2, 3, 4, 5, and 10 years) and collection of nasopharyngeal samples. Parents also kept a daily symptom diary for symptoms of respiratory infection in their child. The data extracted from these samples and measurements covered multiple 'domains' of asthma pathogenesis, including respiratory infection, allergen sensitisation, and clinical or demographic background.

## Measurements in CAS

For each child and visit, the investigators of CAS recorded metrics related to suspected or known modulators of asthma risk. These included markers of immune function: (1) IgG, IgG4, and IgE Phadiatop ImmunoCAP antibodies (ThermoFisher, Uppsala, Sweden), covering common allergens such as house-dust mite (HDM, *Dermatophagoides pteronyssinus*), mould, couch grass, ryegrass, peanut, cat dander; (2) IgE and IgG4 Phadiatop Infant and Adult assays (ThermoFisher, Uppsala, Sweden) that target multiple allergens simultaneously (*Ballardini et al., 2006*); (3) skin prick or sensitisation tests (SPT) for HDM, mould, ryegrass, cat, peanut, cow's milk and hen's egg; and (4) cytokine responses (IL-4,5,9,13,10, IFN-γ) following in vitro stimulation of extracted peripheral blood mononuclear cells (PBMCs) by multiple antigen and allergen stimuli, including phytohaemaglutinin (PHA), HDM, cat, peanut and ovalbumin (*Hollams et al., 2009*; *Holt et al., 2010*).

In addition, nasopharyngeal samples (swabs or aspirates, NPAs) were taken from each child during healthy routine visits (healthy samples), and unscheduled visits where parents presented with their child if they have a suspected respiratory infection (disease samples). Frequency and severity of respiratory infections were measured accordingly. NPAs were then screened for viral and bacterial pathogens using rtPCR and 16 s rRNA amplicon sequencing with Illumina MiSeq (San Diego, US), respectively (*Teo et al., 2015*). These NPAs had previously classified by *Teo et al. (2015)*; *Teo et al. (2017)*, based on clustering of bacterial composition, into microbiome profile groups (MPGs) that were associated with healthy respiratory states (health-associated MPGs, for example *Alloiococcus*-, *Staphylococcus*- or *Corynebacterium*-dominated) or infectious respiratory states (infection-associated MPGs, for example *Moraxella*-, *Haemophilus*-, or *Streptococcus*-dominated).

Other collected data included: sex, height and weight; paternal and maternal history of atopic disease; blood levels of basophils, plasmacytoid and myeloid dendritic cells as measured by

fluorescence-assisted cell sorting (FACS); and levels of vitamin D (25-hydroxycholecalciferol, 25(OH) D) (*Hollams et al., 2017*).

## Identification of latent clusters and selection of clustering features

We adopted an exploratory approach to cluster analysis, whereby we attempted to interrogate as much of the existing dataset as possible, identifying latent clusters that arise from the underlying data structure of CAS. We then assessed how these latent clusters correlate with risk of asthma or other markers of pathophysiology, such as degree of allergic sensitisation. All data processing and analysis were done in R v3.3.1 (RRID:SCR_001905). A graphical overview of the analytic process is displayed in *Figure 1—figure supplement 3*.

To identify latent clusters, we applied non-parametric expectation-maximisation ('npEM') mixture modelling to our discovery cohort CAS, using functions from the R package 'mixtools' (*Benaglia et al., 2009a*). This method assumes that frequency distributions of each cluster can be represented by non-parametric density estimates learned from the data in an iterative process. npEM was used because: (1) it was plausible to consider a population as a mixture of subpopulations each with their own distributions; (2) it had advantages over other unsupervised approaches (*Tan et al., 2005*) – for example, with LCA, continuous variables cannot be handled appropriately; with hierarchical clustering, poor decisions made early in the classifying process are not easily amended; (3) many variables were categorical or non-Gaussian, so theoretically a non-parametric approach should be superior to a Gaussian mixture model or k-means approach; and (4) inherent within mixture models is an intuitive method for supervised classification of other datasets into similar clusters.

We used a largely non-selective approach to choosing features for cluster analysis, in that we attempted to retain as many CAS individuals and variables as possible. However, we did enforce certain quality-control measures such as excluding variables ('features') that had missing data for >20% subjects (442 variables removed), and subjects with missing data for >30% of the remaining variables (39 subjects removed). Also excluded were features pertaining to our primary outcomes of interest: incidence of parent-reported wheeze, physician-diagnosed asthma and hayfever at all timepoints. We specifically excluded these from feature selection so we could determine how subsequent clusters differ in these outcomes, even when clustering was not explicitly driven by them. On the other hand, eczema was not excluded because of evidence that infantile eczema may itself influence the risk for subsequent sensitisation and asthma (*Gustafsson et al., 2000*). Frequency of wheeze in the context of respiratory infection was also included, as it was a symptomatic marker of infection severity. Variable reduction resulted in $M$ = 174 variables remaining out of an original 659. The complete list of variables included as clustering features is provided in *Supplementary file 1* – table supplement 1, and importantly covers multiple domains including demographic (family history of atopy, household size), clinical (incidence of childhood eczema), immunological (IgE, IgG, IgG4, SPT) and microbiological (respiratory infections, viral pathogens associated with infection) features. By virtue of study design and quality control measures, many of the clustering features were related to immunological function or respiratory infection in the first 3 years of life.

Highly skewed features, such as antibody and cytokine levels, were subjected to logarithmic (base 10) transformation. We also applied limited thresholding to some variables (cytokine responses, antibody assays), based on best practice for the reported limit-of-detection (LOD) of the measuring devices. The LOD for IgE was 0.03 kU/L; for IgG4, 0.0003 µg/L; for IgG, 0.4 mg/L. For these variables, we assigned any values below the LOD to half the LOD (i.e. 0.015 kU/L, 0.00015 µg/L, and 0.2 mg/L, respectively). For stimulated cytokine expression above unstimulated control, any zero or negative values (i.e. unstimulated control had equal, or greater, expression than stimulated), were converted to 0.000001 units or 0.01 pg/ml for mRNA and protein variables, respectively. Positional standardisation scaling was then applied across all variables, to equally weight the contributions of each feature to the mixture model. This involved replacing each value $x_{ij}$ for individual $i$ of feature $j$, by:

$$\frac{x_{ij} - \mathrm{med}(x_j)}{\max(x_j) - \min(x_j)}$$

where functions med, max and min refer to the median, maximum, and minimum for the complete-case dataset for feature $j$, respectively.

## Cluster analysis using non-parametric mixture modelling

The processed and scaled CAS dataset was further split into those subjects with no missingness in the remaining variables ('complete-case', 186 subjects, 174 variables); versus those who had limited missingness of <30% variables ('low-missingness', 36 subjects, 174 variables). Cluster analysis was performed initially in the complete-case CAS subset to generate an npEM model.

The mathematical theory underpinning npEM has already been described extensively in other sources (*Benaglia et al., 2009b*). In brief, it involves three steps: (1) an expectation or E-step, which calculates the posterior probability of membership in cluster $k$, given the observed dataset, estimated mixing proportions $\lambda_k$, and probability distribution for $k$; (2) a maximisation or M-step, which calculates the mixing proportions $\lambda_k$ from the cluster memberships determined above; (3) a non-parametric kernel density estimation step, which calculates the probability distribution based on a kernel density function for each cluster $k$ and clustering feature $j$. These steps were then iterated until the model converged to a point where log-likelihood values were maximised.

As with any EM algorithm, an initial state must first be set prior to commencing the iterative process. To do this, we used a constant seed state ('set.seed(1)') to allow reproducibility of results. Based on these pseudo-random centroids for a set number of clusters $L$, the initial state was then determined by k-means clustering as in Benaglia et al (*Benaglia et al., 2009b*). The other options in npEM were set to defaults. These included the use of non-stochastic (deterministic) as opposed to a stochastic method; the use of a standard normal density function as the kernel function; and the use of default constant bandwidths for estimating kernel densities (*Benaglia et al., 2009b*).

The ideal number of clusters $L$ was determined by two methods. Firstly, we performed hierarchical clustering on the complete-case dataset, and scrutinised the dendrogram as well as a scree plot for an optimal cut-off using the 'knee method' (*Tan et al., 2005*). We observed that this occurred at around $L$ = 3 or 4. Secondly, we repeated npEM clustering for values of $L$ = 1,2,...,20, and calculated the Bayesian information criterion (BIC) for each of these, using the formula:

$$\mathrm{BIC} = -2\log(\hat{p}) + \nu \log(N)$$

where $P$ is the maximum likelihood, $\nu = L \times M + (L - 1)$, and $L$, $M$, $N$ are total number of clusters, clustering features, and individuals respectively. The optimal number of clusters was again determined to be around $L$ = 3 or 4, based on minimum BIC observed. For the sake of parsimony, we set the number of clusters to three.

## Classification of test datasets using mixture model densities

The density functions generated by the resultant npEM model were then used to classify as many subjects of the low-missingness subset as possible. This method relied on the assumption that distributions observed in the 'training' (complete-case) dataset were representative of distributions that existed in 'test' (low-missingness or external) datasets.

Classification was performed as follows: consider individual $i$ of $N$; clustering feature or coordinate $j$ of $M$; and component or cluster $k$ of $L$. For each individual $i$ belonging to known cluster $k=K$, let the kernel density function for coordinate $j$ be $f_{jK}(x_{ij})$. We now assume that the coordinates $j$ were independent of each other. Although this was not truly the case – for instance, weak correlation exists between IgE and IgG4 of different allergen specificities in the CAS dataset [23] – we believed the assumption was justified given our theory-naive and exploratory approach. With this assumption, the joint distribution for individual $i$ in cluster $K$ should be the product of density functions for all $j$ given $K$. and therefore the probability of individual with value $x_{ij}$ belonging to cluster $K$ was:

$$P(k = K | x_{ij}) = \frac{\lambda_K \prod_{j=1}^{M} f_{jK}(x_{ij})}{\sum_{k=1}^{L} \lambda_k \prod_{j=1}^{M} f_{jk}(x_{ij})}$$

In addition to this, we made two other assumptions: (1) if $x_{ij}$ was missing, then the density was assumed to be one, or $f_{jK}(x_{ij}) = 1$; (2) else, if $x_{ij} <$ min($x_j$), the minimum value in feature $j$ for which

there was a non-zero density value, then the density was equal to that of the minimum value, that is $f_{jK}(x_{ij}) = f_{jK}(\min(x_j))$. Likewise, if $x_{ij} > \max(x_j)$, then $f_{jK}(x_{ij}) = f_{jK}(\max(x_j))$.

Individuals with membership probability greater than 90% for cluster $K$ were classified into $K$. Using this method, an additional 31 individuals from 36 were successfully classified into one of three clusters, for a total combined dataset of 217 classified individuals in CAS.

Finally, we formally defined each CAS cluster using the composite of complete-case and low-missingness datasets, and described each cluster in terms of key characteristics and significant cluster-specific predictors for age-5 wheeze. Importantly, variables that were initially excluded from feature selection were treated as subsequent outcomes for post-hoc comparison of clusters.

## Replication cohorts

The study designs and measurements for the two replication cohorts – the Manchester Asthma and Allergy Study (MAAS) ($N$ = 1085) from Manchester, UK, and the Childhood Origins of Asthma Study (COAST) ($N$ = 289) from Wisconsin, USA – have been described elsewhere (*Belgrave et al., 2014*; *Gern et al., 2002*; *NAC Manchester Asthma and Allergy Study Group et al., 2002*; *Lemanske, 2002*). COAST, like CAS, was comprised of high-risk individuals with a known family history of asthma or allergy; while MAAS included individuals without family history.

In terms of matching variables for replication, all cohorts had measurements that covered the three major 'domains' of asthma pathogenesis: respiratory infection, allergen sensitisation, and clinical or demographic background. COAST had a comprehensive collection of respiratory infection and IgE-type measurements, but no IgG4 measurements. MAAS had multiple measurements of IgE and SPT-type variables. Following consultation with investigators from all three cohorts, clustering features were matched based on proximity of timepoint and phenotype. Respiratory infection phenotypes (ARI, LRI, URI, fLRI, wLRI) were generated in COAST and MAAS using recorded data, to approximate CAS infection phenotypes as closely as possible. Specifically, LRI was defined as respiratory infection with evidence of lower respiratory tract involvement in the form of chest sounds (wheeze, rattle, whistle), or increased respiratory effort (retractions, tachypnea, cyanosis); URI was defined as a cold-like infection limited to the upper respiratory tract, without signs of LRI. IgE and IgG4 assays for MAAS and COAST were performed using ImmunoCAP and UniCAP, respectively. Both replication cohorts recorded basic demographic data, and exposures to pets, childcare, and tobacco smoke. The complete list of clustering features and the matching scheme across cohorts is provided in *Supplementary file 1* – table supplement 1.

The npEM clusters were described and validated in MAAS and COAST. This replication was performed by applying the density-function-derived classifier used previously for the low-missingness CAS subjects. Because these external cohorts did not necessarily share the same clustering features or variables as CAS (*Supplementary file 1* – table supplement 1), we assumed that the respective densities for these variables were $f_{jK}(x_{ij}) = 1$ for the $j^{th}$ feature and $K^{th}$ cluster. In doing so, this was effectively the same as using a model where the missing features were excluded, and only those features common to both CAS and MAAS (or COAST) were used; or equivalently, where we assumed that each member of MAAS or COAST was missing values in those particular features. Because these 'CAS-derived' npEM models were non-identical to the original npEM models in CAS, we tested whether 'MAAS-like' and 'COAST-like' algorithms (CAS-derived model as applied to MAAS or COAST, respectively) generated similar clusters to the original CAS clusters, when applied back onto CAS (Results).

## Cluster validity and stability

Internal validation of the clusters in the complete-case CAS dataset was performed by use of silhouette widths. Briefly, we calculated the silhouette widths for each cluster as per *Rousseeuw (1987)*. For an individual, the closer the silhouette width is to one, the more appropriate the cluster membership; while the closer it is to negative one, the more likely it has been misclassified.

Cluster stability was assessed by performing leave-one-out (LOO) analysis – that is, we applied the npEM algorithm to a subset of the complete-case dataset – an $N$-1 by $M$ dataset ($N$ = 186, $M$ = 174) for a total of $N$ times, leaving out an individual each time. A similar process was repeated $M$ times on an $N$ by $M$-1 dataset, leaving out one clustering feature at a time. The Jaccard indices for each iteration were then calculated in comparison to known clusters from the original complete-

case *N* by *M* dataset, and averaged across each assigned cluster. Cluster labels for each iteration were assigned based on whichever complete-case cluster yielded the smallest Jaccard index. This whole process was then repeated with 10 random seeds ('set.seed(1)' through to 'set.seed(10)') for determining the initial state for npEM. The final averaged Jaccard indices for each cluster thus represented the mean stability of each cluster.

## Decision tree analysis

Decision tree analysis was performed using a number of different partitioning schemes. Classification trees with recursive partitioning were built from CAS clusters using the R package 'rpart' (*Therneau and Atkinson, 2015*), an open-source implementation of CART. The motivation for decision trees was to identify the variables that most *strongly separated* the clusters and wheezing status, and not necessarily variables that were most predictive.

For tree outcomes (end-nodes), we investigated both cluster membership and presence of age-5 wheeze given cluster membership. That is, decision trees were generated to identify the biological features that most strongly distinguished each npEM cluster ('Simple Tree'), as well as npEM cluster ×age-5 wheeze status ('Comprehensive Tree').

We used two different schemes for selecting predictors on which to base the partitions: 1) include all predictors that were used as clustering features in the original npEM model; 2) include only predictors from one timepoint (variables from age 6 m, 1, 2 or 3). The motivation for the latter was that we wanted to see whether measurements taken at a specific timepoint in early infancy could strongly distinguish between clusters. For the former scheme, we excluded all age-5 features related to wheeze (e.g. LRIs, wheezy LRIs at age 5) as decision nodes, because of definitional overlap with our primary outcome of interest (age-5 wheeze).

Decision trees were then pruned based on the complexity parameter that minimised cross-validated error. Final classification into tree clusters was manually performed based on the pruned tree, and not by automatic classification using the 'predict' function for the 'rpart' tree object – this was because, for the latter, individuals who are missing key variables were re-classified based on the next best, non-missing, surrogate variable (*Therneau and Atkinson, 2015*). Thus, it resulted in children being erroneously classified into a tree cluster even when they were missing key classifier variables.

The decision tree analyses generated thresholds which were then compared with existing thresholds for atopy (any specific IgE at age 2 $\geq$ 0.35 kU/L, and/or any specific SPT at age 2 $\geq$ 2 mm) (*Frith et al., 2011*) in terms of predicting disease outcomes of interest.

## Statistical analyses

We performed statistical analyses comparing clusters in terms of multiple variables, especially those not used as clustering features. Of interest to us were the primary outcomes of asthma diagnosis and parent-reported wheeze at each timepoint. Where appropriate, we used *t*-tests, Mann-Whitney-Wilcoxon tests, ANOVAs, Kruskal-Wallis tests, chi-squared and Fisher exact tests; and logistic and linear regression. For summary statistics, multiple testing adjustment was performed using the Benjamini-Yekutieli (BY) method, for all across-cluster tests (Cluster $\times$ trait); and for all comparisons between clusters (CAS1 vs. 2, 1 vs. 3, and 2 vs. 3). The BY method was chosen as it accounted for positive dependency across the highly correlated variables in the CAS dataset (*Benjamini and Yekutieli, 2001*). For variables that underwent logarithmic transformation for statistical analysis, we used geometric mean to describe central tendency.

We then determined the predictors for age-5 wheeze within each cluster. Repeated-measures ANOVAs were performed for selected predictors of age-5 wheeze. For each potential predictor, generalised linear regression models (GLMs) were generated with and without a base set of covariates (sex, family history of asthma, BMI where available). The pool of variables found to be statistically significant (at least p<0.05) in the above analyses were further restricted, such that strongly collinear predictors were avoided, and at most one timepoint was considered for each predictor type. Targeted multiple regression models were then built by selecting predictors from this constrained pool. Stepwise backward elimination was applied, in which the predictor with the largest p-value was eliminated at each step, until all remaining predictors have significant p<0.05.

Using the 'lrtest' function from the R package 'Epidisplay' (*Chongsuvivatwong, 2015*), likelihood ratios were examined to check how much cluster membership or classification improved upon prediction of age-5 wheeze compared to traditional makers of atopy.

## Additional information

### Funding

| Funder | Grant reference number | Author |
| --- | --- | --- |
| National Health and Medical Research Council | 1049539 | Michael Inouye |
| National Health and Medical Research Council | PhD Scholarship | Howard HF Tang |

The funders had no role in study design, data collection and interpretation, or the decision to submit the work for publication.

### Author contributions

Howard HF Tang, Conceptualization, Data curation, Formal analysis, Validation, Investigation, Visualization, Methodology, Writing—original draft, Writing—review and editing; Shu Mei Teo, Data curation, Formal analysis, Methodology, Writing—review and editing; Danielle CM Belgrave, Resources, Software, Formal analysis, Validation, Methodology, Writing—review and editing; Michael D Evans, James E Gern, Resources, Data curation, Formal analysis, Validation, Methodology, Writing—review and editing; Daniel J Jackson, Robert F Lemanske, Resources, Data curation, Validation, Methodology, Writing—review and editing; Marta Brozynska, Conceptualization, Resources, Data curation, Formal analysis, Validation, Investigation, Visualization, Methodology, Writing—original draft, Project administration, Writing—review and editing; Merci MH Kusel, Resources, Data curation, Formal analysis, Methodology, Writing—review and editing; Sebastian L Johnston, Resources, Data curation, Software, Formal analysis, Funding acquisition, Validation, Methodology, Writing—review and editing; Angela Simpson, Resources, Validation, Methodology, Writing—review and editing; Adnan Custovic, Resources, Data curation, Validation, Writing—review and editing; Peter D Sly, Conceptualization, Resources, Data curation, Validation, Writing—original draft, Project administration, Writing—review and editing; Patrick G Holt, Conceptualization, Resources, Data curation, Supervision, Methodology, Writing—original draft, Writing—review and editing; Kathryn E Holt, Conceptualization, Resources, Formal analysis, Supervision, Funding acquisition, Validation, Methodology, Writing—original draft, Writing—review and editing; Michael Inouye, Conceptualization, Formal analysis, Supervision, Funding acquisition, Methodology, Writing—original draft, Project administration, Writing—review and editing

### Author ORCIDs

Howard HF Tang http://orcid.org/0000-0001-6422-0270
Michael D Evans http://orcid.org/0000-0001-7449-3993
Adnan Custovic http://orcid.org/0000-0001-5218-7071
Michael Inouye http://orcid.org/0000-0001-9413-6520

### Ethics

Human subjects: Ethics approval and consent requirements for each cohort were met as follows: The CAS study was approved by the ethics committees of the King Edward Memorial and Princess Margaret Hospitals in Western Australia; fully informed parental consent was obtained for all subjects. The COAST study was approved by the Human Subjects Committee of the University of Wisconsin. The MAAS study was approved by a Manchester Local Research Ethics Committee (ERP/94/032; SOU/00/258; 03/SM/400; Study registration ISRCTN72673620); fully informed parental consent was obtained for all subjects across all cohorts.

Decision letter and Author response
Decision letter https://doi.org/10.7554/eLife.35856.034
Author response https://doi.org/10.7554/eLife.35856.035

## Additional files

### Supplementary files

• Supplementary file 1. All table supplements
DOI: https://doi.org/10.7554/eLife.35856.025

• Supplementary file 2. Comparison of variables (respiratory, immunological, clinical) across CAS clusters. Analogous to Table Supplement 3.
DOI: https://doi.org/10.7554/eLife.35856.026

• Supplementary file 3. Predictors for age-five wheeze within each CAS cluster, with demographic covariates (sex, BMI, parental history of asthma). Analogous to Table Supplement 7.
DOI: https://doi.org/10.7554/eLife.35856.027

• Transparent reporting form
DOI: https://doi.org/10.7554/eLife.35856.028

### Data availability

This study utilises extensive data from human subjects, specifically paediatric cohorts, for which eLife's policies recognise that there can be strong reasons to restrict access. For each of the cohorts involved in our study (CAS, COAST, MAAS), parents were consented on the use of biomedical data for allergy and asthma research, but not for the open sharing of their or their children's data. Studies were run in the late 1990s and early 2000s and we do not have ethics permission to attempt to recontact families to seek consent. Importantly, we note that key data features could risk re-identification of subjects (e.g. demographic data from small communities). However, we have provided public data at the summary level which can be used for subsequent studies, such as replication and meta-analysis. This is standard practice in sensitive data settings, such as genome-wide association studies. These data have been uploaded as Excel spreadsheets to FigShare for ease of data extraction: **Supplementary Table 3** https://figshare.com/articles/Supplementary_File_1_1/6934052; **Supplementary Table 7** https://figshare.com/articles/Supplementary_File_1_2/6934055

The following datasets were generated:

| Author(s) | Year | Dataset title | Dataset URL | Database and Identifier |
|---|---|---|---|---|
| Howard HF Tang, Michael Inouye, Kathryn E Holt | 2018 | Data from supplementary table 3 | https://figshare.com/articles/Supplementary_File_1_1/6934052 | Figshare, Supplementary_File_1_1/6934052 |
| Howard HF Tang, Michael Inouye, Kathryn E Holt | 2018 | Data from supplementary table 7 | https://figshare.com/articles/Supplementary_File_1_2/6934055 | Figshare, Supplementary_File_1_2/6934055 |

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
