## [Decision Letter]

Thank you for submitting your article "Non-parametric mixture models identify trajectories of childhood immune development relevant to asthma and allergy" for consideration by *eLife*. Your article has been reviewed by three peer reviewers, including M Dawn Teare as the Reviewing Editor and Reviewer #1, and the evaluation has been overseen Arup Chakraborty as the Senior Editor. The following individual involved in review of your submission has agreed to reveal their identity: Spyridon Megremis (Reviewer #3).

The reviewers have discussed the reviews with one another and the Reviewing Editor has drafted this decision to help you prepare a revised submission.

Summary:

This is a comprehensive piece of research. Several rich datasets with longitudinal measurements on young children at risk of asthma have been analysed using unsupervised learning methods to discover clusters. This appears to be novel research for the field and has uncovered three clusters of trajectories which have been validated to some extent. This is a robust, well-described study representing a modern approach to identify key events in early-life that might predispose, or even define, the "fate" of children who develop (or not) wheeze atopy and asthma. Conceptually, the study is based on three major realizations: (1) the dynamic state of biological systems (use of longitudinal data), (2) the immune status of the host (immune related variables), and (3) the exposure to our microbial partners (environment).

We request that the authors change the emphasis of the paper, remove some of the later statistical analyses, provide more detail on the replication cohorts and focus on the implications for asthma rather than the nuts and bolts of how the methods employed work.

Essential revisions:

1) There should be more of a focus on the translational findings. Compared to an extremely long Results section, the Discussion is not even one page and the first paragraph is a summary of the analysis pipeline. Remove the non-parametric mixture models from the title, and adjust the main body of the manuscript to the translational findings with most of the space focusing on atopy and microbial exposure. The authors have identified three groups of children: (1) CAS1 is a low-risk cluster with a decline in wheeze rates by age 10. No Th2 profile was found and rates of allergen stimulation were minimal. Frequency of respiratory infection was low, but importantly, high LRI frequency and wLRI was a risk factor for age 5 wheeze. Backward elimination showed that age-3 wLRI, age-4 LRI frequency and certain bacterial species were significant while LRI and wLRI in the first 3 years were predictors for age-5 wheeze. (2) CAS2 is a low-risk but allergy susceptible. Again there is a drop in wheeze rates. Predictive factors for wheeze included LRI, wheezy LRI, fLRI and IgE but with no interaction between microbial exposure and atopy. (3) CAS3 is a high risk atopic cluster with persistent wheeze, elevated IgE, IgG and IgG4 to allergens and severe LRIs in early-life. Use a figure to summarise the results as described in subsections “CAS1: low-risk, non-atopic cluster with transient wheeze”, “CAS2: low-risk cluster susceptible to atopic and non-atopic wheeze” and “CAS3: high-risk atopic cluster with persistent wheeze”. These are the most important findings because they demonstrate that in the absence of a severe pathological phenotype the respiratory system is "open" to external stimuli that can truly act in a causative manner and lead to (or not) disease. For example, CAS2 had fewer siblings than CAS1 (evidence of biodiversity effect). In presence of atopy, most probably the tissue biome (host and microbiome) is "fixed" and determines the trajectory of the individual.

2) The use of decision tree analysis should be described entirely in the Materials and methods section. It accounts for almost a page in the manuscript and this space may be used in the discussion part to comment on the earlier findings. This would free up a bit more space to clarify why the cascade of methods was used. For example, it is not immediately obvious why within each cluster a step-wise logistic regression predictive model is built to relate the cluster features with the risk of asthma. There are 174 features being used to identify the clusters and each cluster will contain few children so this approach feels the weakest part of the analysis. The manuscript comes with many tables and a very extensive set of supplementary materials. It is difficult to see the flow of how each step has been done and why. These issues need to be clarified.

3) The authors use good machine learning; however, given the visualization, we strongly suspect that simpler methods would also work well. The authors should perform a PCA analysis to add a figure to see the clusters on a PCA plot post inverse normalization – it should be strong given Figure 1. The impact of the paper rests on the impact on asthma, not on the cleverness of the machine learning, so if simpler methods also confirm a strong effect this shows that the finding is robust. If not, then it points to the need for the more sophisticated method. Either result is interesting.

4) A big concern is the replication. The authors used other cohorts, but there is no obvious discussion or presentation in the main text about which variables matched between cohorts and how the between cohort variable matching was made. The supplement has far more (Table 1 and the flow diagram) but not the supporting information about how this works. More details are needed here.

---

## [Author Response]

Essential revisions:1) There should be more of a focus on the translational findings. Compared to an extremely long Results section, the Discussion is not even one page and the first paragraph is a summary of the analysis pipeline. Remove the non-parametric mixture models from the title, and adjust the main body of the manuscript to the translational findings with most of the space focusing on atopy and microbial exposure. The authors have identified three groups of children: (1) CAS1 is a low-risk cluster with a decline in wheeze rates by age 10. No Th2 profile was found and rates of allergen stimulation were minimal. Frequency of respiratory infection was low, but importantly, high LRI frequency and wLRI was a risk factor for age 5 wheeze. Backward elimination showed that age-3 wLRI, age-4 LRI frequency and certain bacterial species were significant while LRI and wLRI in the first 3 years were predictors for age-5 wheeze. (2) CAS2 is a low-risk but allergy susceptible. Again there is a drop in wheeze rates. Predictive factors for wheeze included LRI, wheezy LRI, fLRI and IgE but with no interaction between microbial exposure and atopy. (3) CAS3 is a high risk atopic cluster with persistent wheeze, elevated IgE, IgG and IgG4 to allergens and severe LRIs in early-life. Use a figure to summarise the results as described in subsections “CAS1: low-risk, non-atopic cluster with transient wheeze”, “CAS2: low-risk cluster susceptible to atopic and non-atopic wheeze” and “CAS3: high-risk atopic cluster with persistent wheeze”. These are the most important findings because they demonstrate that in the absence of a severe pathological phenotype the respiratory system is "open" to external stimuli that can truly act in a causative manner and lead to (or not) disease. For example, CAS2 had fewer siblings than CAS1 (evidence of biodiversity effect). In presence of atopy, most probably the tissue biome (host and microbiome) is "fixed" and determines the trajectory of the individual.

We agree that the manuscript would benefit from emphasis on the translational findings.As suggested, we have now made extensive changes to the main text to emphasise the biological and translational aspects, particularly in the Discussion.

Most of the Discussion has been revised. Relevant commentary which was previously in the supplementary materials has been moved to the main text (Discussion section). We discuss implications for pathophysiology and clinical definitions of atopy (Materials and methods section), as well as possible roles of respiratory infections and the microbiome.

The title has been changed as requested, to “Trajectories of childhood immune development and respiratory health relevant to asthma and allergy”.

A graphical summary of our clusters as described in the Discussion (subsections “Cluster 3 is a high-risk, multi-sensitised, atopic phenotype”, “Role of early-life HDM hypersensitivity”, “Role of early-life food and peanut sensitization”) was previously provided as Figure 7; however to make this more useful for the reader, we have now made this summary Figure 2.

2) The use of decision tree analysis should be described entirely in the Materials and methods section. It accounts for almost a page in the manuscript and this space may be used in the discussion part to comment on the earlier findings. This would free up a bit more space to clarify why the cascade of methods was used. For example, it is not immediately obvious why within each cluster a step-wise logistic regression predictive model is built to relate the cluster features with the risk of asthma. There are 174 features being used to identify the clusters and each cluster will contain few children so this approach feels the weakest part of the analysis. The manuscript comes with many tables and a very extensive set of supplementary materials. It is difficult to see the flow of how each step has been done and why. These issues need to be clarified.

To establish better flow and avoid redundancy, we have moved the methods to between Introduction and Results. Tables 2, 3 and 4in the main text have been reduced to include only the most important findings that we wish to highlight. The complete versions of these tables remain in Supplementary file 1.

For the decision tree analysis, we have unified and condensed the relevant methods and supplementary methods into one section, and likewise for results and supplementary results. Also, we have moved all these sections on decision tree analysis to Supplementary file 1 and Figure 5—figure supplements 1, 2 and 3, to make room for other methods and discussion.

Stepwise elimination was used to identify and narrow down predictors for *age-five wheeze within each cluster*. This is distinct from using features to determine cluster membership, which was instead done with the npEM method. The methods on regression analysis with stepwise elimination have been expanded to include the following:

“For each potential predictor, generalised linear regression models (GLMs) were generated with and without a base set of covariates (sex, family history of asthma, BMI where available). […] Stepwise backward elimination was applied, in which the predictor with the largest p-value was eliminated at each step, until all remaining predictors have significant *p*<0.05.”

The “univariate” GLMs demonstrated that multiple predictors were significant within CAS1 and CAS2, even with multiple testing adjustment. Modelling multiple relevant predictors together (rather than individually) is important because we want to see whether certain predictors act independently or interactively with each other. We stress that we were selective with the variables considered for predictor testing. Most importantly, we were able to replicate the CAS-identified predictors in an external cohort, COAST (subsection “External replication of clusters in MAAS and COAST” and Figure 5—figure supplement 4).

3) The authors use good machine learning; however, given the visualization, we strongly suspect that simpler methods would also work well. The authors should perform a PCA analysis to add a figure to see the clusters on a PCA plot post inverse normalization – it should be strong given Figure 1. The impact of the paper rests on the impact on asthma, not on the cleverness of the machine learning, so if simpler methods also confirm a strong effect this shows that the finding is robust. If not, then it points to the need for the more sophisticated method. Either result is interesting.

We agree that simpler analysis methods should be preferred over complex methods, however the selection of method must be statistically appropriate. PCA is a very common data exploration method but importantly (i) it is inappropriate for heterogeneous data types (i.e. datasets which include both continuous and discrete/binary variables); and (ii) it is not a principled approach to identify clusters on its own. As such, it was necessary from the outset of our study to use a statistically rigorous method which explicitly identified clusters and was non-parametric.

However, we do believe that visualisation of the top principal components should *broadly* recapitulate the clusters we have identified here using the non-parametric mixture model. We performed PCA with the complete-case CAS data (N=186), and below is a scatterplot showing the first two principal components (PCs) which account for only 16.7% of total variance. We observe that points from the same npEM cluster are generally close together especially across PC1 (left-to-right).

To address the reviewer's concerns, we have added this PCA to paragraph 1 of the Results and Figure 1—figure supplement 1, while also noting that the npEM approach offers the advantage of explicitly identifying clusters and handling heterogeneous data.

4) A big concern is the replication. The authors used other cohorts, but there is no obvious discussion or presentation in the main text about which variables matched between cohorts and how the between cohort variable matching was made. The supplement has far more (Table 1 and the flow diagram) but not the supporting information about how this works. More details are needed here.

We have added additional information on how variable matching was performed across cohorts (Materials and methods), as follows:

“In terms of matching variables for replication, all cohorts had measurements that covered the three major “domains” of asthma pathogenesis: respiratory infection, allergen sensitisation, and clinical or demographic background. […] The complete list of clustering features and the matching scheme across cohorts is provided in Supplementary file 1.”